# Electrochemical-repaired porous graphene membranes for precise ion-ion separation

Zongyao Zhou[1,2], Kangning Zhao [1], Heng-Yu Chi[1], Yueqing Shen[1], Shuqing Song [1], Kuang-Jung Hsu[1], Mojtaba Chevalier [1], Wenxiong Shi [3] & Kumar Varoon Agrawal [1]✉

The preparation of atom-thick porous lattice hosting Å-scale pores is attractive to achieve a large ion-ion selectivity in combination with a large ion flux. Graphene film is an ideal selective layer for this if high-precision pores can be incorporated, however, it is challenging to avoid larger non-selective pores at the tail-end of the pore size distribution which reduces ion-ion selectivity. Herein, we develop a strategy to overcome this challenge using an electrochemical repair strategy that successfully masks larger pores in large-area graphene. 10-nm-thick electropolymerized conjugated microporous polymer (CMP) layer is successfully deposited on graphene, thanks to a strong π-π interaction in these two materials. While the CMP layer itself is not selective, it effectively masks graphene pores, leading to a large $Li^+/Mg^{2+}$ selectivity from zero-dimensional pores reaching 300 with a high $Li^+$ ion permeation rate surpassing the performance of reported materials for ion-ion separation. Overall, this scalable repair strategy enables the fabrication of monolayer graphene membranes with customizable pore sizes, limiting the contribution of nonselective pores, and offering graphene membranes a versatile platform for a broad spectrum of challenging separations.

Membranes with the ability to separate ions and small molecules with high selectivity and a high flux hold the potential to revolutionize the energy, water, and chemical sectors, making them essential for advancing the sustainability of society[1–5]. The advancement of the field of membranes relies on the exploration of new materials and novel membrane fabrication methods. With the commercialization of chemical vapor deposition (CVD) graphene produced by the roll-to-roll technique[6–9], monolayer graphene has an increasingly attractive potential as a practical membrane material[10–15]. Its atomic-thin lattice is the thinnest possible molecular barrier. Combined with the possibility of uniform Å-scale pores, porous graphene has the potential to yield ultimate separation performance[16–20]. Nonetheless, avoiding non-selective pores in monolayer graphene remains a challenge[18,21,22]. A significant number of large non-selective pores inevitably occur when introducing pores in monolayer graphene, whether through a direct bottom-up synthesis of crystalline nanoporous graphene[23] or the more commonly used post-synthetic etching[24,25]. These large pores, combined with any cracks and tears during the membrane fabrication process, compromise the selectivity of the resulting membranes[16,26].

Using a masking layer to negate the effect of non-selective large pores and cracks in graphene membranes provides a promising strategy for fabricating highly selective graphene membranes. For instance, a masking layer with Å-scale apertures would mitigate the rapid, non-selective transport arising from the nanoscale tears, cracks, and large non-selective pores in graphene, thereby enhancing selectivity[27–29]. Recently, polyelectrolyte coating[30] and interfacial polymerization[14,31] have been adopted to mask and seal the defects on graphene membranes. These endeavors have demonstrated notable

[1]Laboratory of Advanced Separations (LAS), École Polytechnique Fédérale de Lausanne (EPFL), Sion CH-1950, Switzerland. [2]State Key Laboratory of Urban Water Resource and Environment, School of Environment, Harbin Institute of Technology, Harbin 150090, P. R. China. [3]Institute for New Energy Materials and Low Carbon Technologies, School of Materials Science and Engineering, Tianjin University of Technology, Tianjin 300387, P. R. China. ✉e-mail: kumar.agrawal@epfl.ch

success in enhancing selectivity. However, the masking layer has to be carefully designed such that the masking layer by itself does not limit the transport. Otherwise, one would lose the advantage of using atom-thick graphene film as the selective layer. Further, the masking layer should have a strong interaction with the selective layer. A poor interaction will result in gaps between graphene and the masking layer, leaving the masking strategy ineffective.

Herein, we report a masking layer for graphene prepared from an ultrathin (~10 nm) conjugated microporous polymer (CMP) film by an electropolymerization technique. The CMP layer hosting a pore window of ~1 nm has a strong interaction with graphene, effectively masking graphene pores, and reducing the role of the tail-end of the pore size distribution (PSD) with pores larger than 1 nm. This enables the repair of PSD and results in a large ion-ion selectivity from Å-scale graphene pores. This bottom-up electrochemical repair approach offers numerous advantages: (i) by utilizing the conductivity of graphene, a CMP mask layer with uniform intrinsic micropores can be easily obtained in situ on the surface of porous graphene, (ii) the thickness of the polymer mask layer on graphene can be precisely controlled at the nanoscale via tuning electrochemical parameters, which is beneficial to limit the thickness of the masking layer to a few nanometers; (iii) electrochemical strategy is gentle and avoids damages to the graphene during the creation of masking layer, and (iv) the graphene film is mechanically reinforced by the CMP layer which prevents graphene from cracking and tearing. Subsequently, the porous graphene membranes and the repaired counterparts were subjected to ion-ion separation experiments. Large $Li^+/Mg^{2+}$ selectivity (reaching 300) accompanying high $Li^+$ flux from the CMP-masked graphene could be achieved from a centimeter-scale single-layer graphene coupon overcoming the performance of the state-of-the-art membranes.

## Results

### Carbon nanotubes supported graphene membrane

Figure 1a shows the schematic of the basic structure of the graphene membranes before introducing the CMP masking layer. Briefly, a large piece of CVD monolayer graphene on Cu foil was used as the starting material[12,32]. A free-standing carbon nanotube (CNT) film with a thickness of 270 nm, pre-prepared by facile filter-coating and substrate dissolution[33,34], was utilized as a mechanically reinforcing porous scaffold to allow tear-free graphene transfer. The CNT film was transferred and dried onto the top of graphene-Cu. After etching the Cu foil, tear-free CNT-supported graphene film could be easily obtained. More details about the preparation and characterizations of the CNT films are shown in Supplementary Figs. 1-7 (Supplementary Notes 1 and 2). Scanning electron microscopy (SEM) of the edge of the film (Fig. 1b) reveals distinguishing contrast from continuous graphene film well supported by a uniform CNT network with an interlocked array of nanotubes. More SEM results are presented in Supplementary Figs. 8 and 9, and discussed in Supplementary Note 3. In the selected area electron diffraction (SAED) analysis, the characteristic patterns of graphene[35,36] (Fig. 1c) were observed alongside the SAED patterns of CNT support. The patterns could be observed on the entire area of film suspended over a transmission electron microscopy (TEM) grid (Supplementary Fig. 10), further signifying the successful transfer and complete coverage of the graphene to the CNT support. This pattern was absent in the control sample without graphene (standalone CNT film, Supplementary Fig. 6). The CNT-reinforced graphene film was

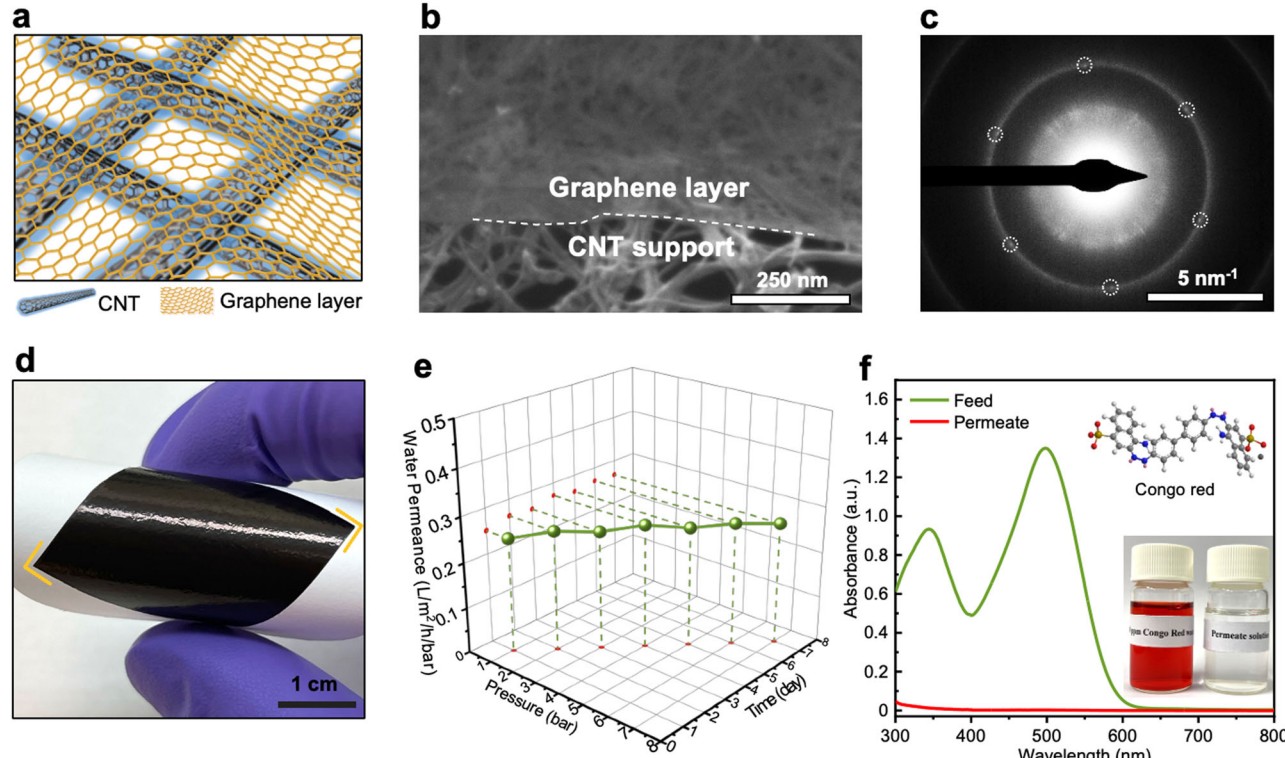

**Fig. 1 | Fabrication of carbon nanotubes (CNT)-supported monolayer graphene membrane. a** Schematic illustration of the membrane structure. The orange honeycombs represent graphene whereas black tubes represent CNT. **b** SEM image of CNT-supported monolayer graphene obtained using a backscattered electron detector. The white dashed line identifies the edge of the upper graphene layer. **c** Selected area electron diffraction (SAED) pattern from the CNT-supported graphene. The pattern from graphene is identified with white circles. **d** Photograph of the CNT-supported graphene membrane. **e** Water permeance of the CNT-supported graphene membrane in a long-term experiment. The graphene in this case is as-synthesized without any intentional pore formation. **f** UV-Vis spectra of the feed and permeate solution consisting of dye (Congo red) dissolved in water. The insets show the molecular structure of the dye on the top and an optical photograph of the feed and permeate solution at the bottom.

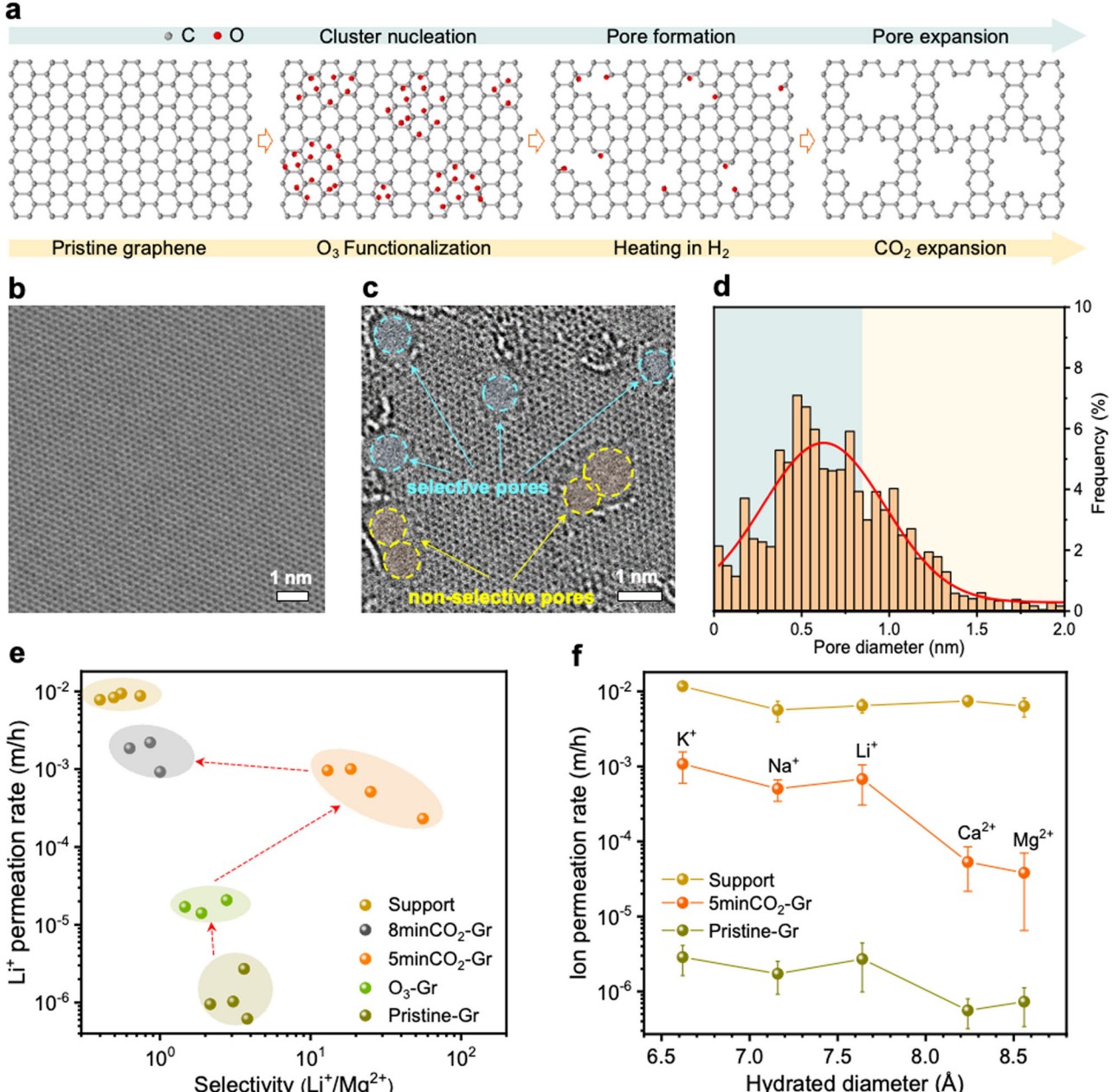

**Fig. 2 | Generation of ion-selective pores in graphene membranes and corresponding ion-ion separation performance. a** Schematic illustration of the strategy for the generation of ion-sieving pores in graphene. Gray and red atoms represent C and O, respectively. **b** Aberration-corrected high-resolution transmission electron microscopy (AC-HRTEM) image of the pristine-Gr. **c** AC-HRTEM image of the 5minCO$_2$-Gr. **d** Pore size distribution from the 5minCO$_2$-Gr where selective and non-selective pores are highlighted in cyan and yellow, respectively. **e** Ion-ion separation performance comparison between the membranes prepared with different pore generation conditions. **f** Ion-ion separation performance of the membranes tested using different ions. The error bar is the standard deviation from at least three samples, and the center of each error bar represents the average data from these samples.

deposited onto a porous polytetrafluoroethylene (PTFE) support (Fig. 1d) for the tests. The resulting membrane was evaluated by its resistance to abrasion with sandpaper[37] (Supplementary Note 4, Supplementary Figs. 11–13), where the membrane showed good mechanical stability, indicating that the adhesion between the layers was strong.

To understand the effectiveness of this transfer approach in preventing cracks and tears, we studied molecular transport through pristine graphene prepared in this way without any intentional defect incorporation. Pristine graphene lattice is dense (Fig. 2b) and impermeable to molecules[38,39]. The only possible transport pathway is through intrinsic multivacancy defects in the polycrystalline graphene

film which are present with a low density[12]. Therefore, a low molecular permeance (e.g., of water) is a good testimony of an effective transfer approach. Indeed, we observed a negligible water permeance (0.26 L/m²/h/bar, Fig. 1e), 99.97% lower than that of the standalone CNT support film (833 L/m²/h/bar, Supplementary Fig. 14), confirming the blockage of water from the pristine graphene film. Notably, this value is more than 2.5-fold smaller than the literature report on water permeance from pristine graphene (0.67 L/m²/h/bar)[10]. This confirms that graphene transferred and supported by CNT film maintains its structural integrity, remaining free from tearing and leakage. In addition, CNT-supported graphene could withstand at least 7 bar pressure in 7-day-long tests (Fig. 1e, Supplementary Figs. 15 and 16), with parity

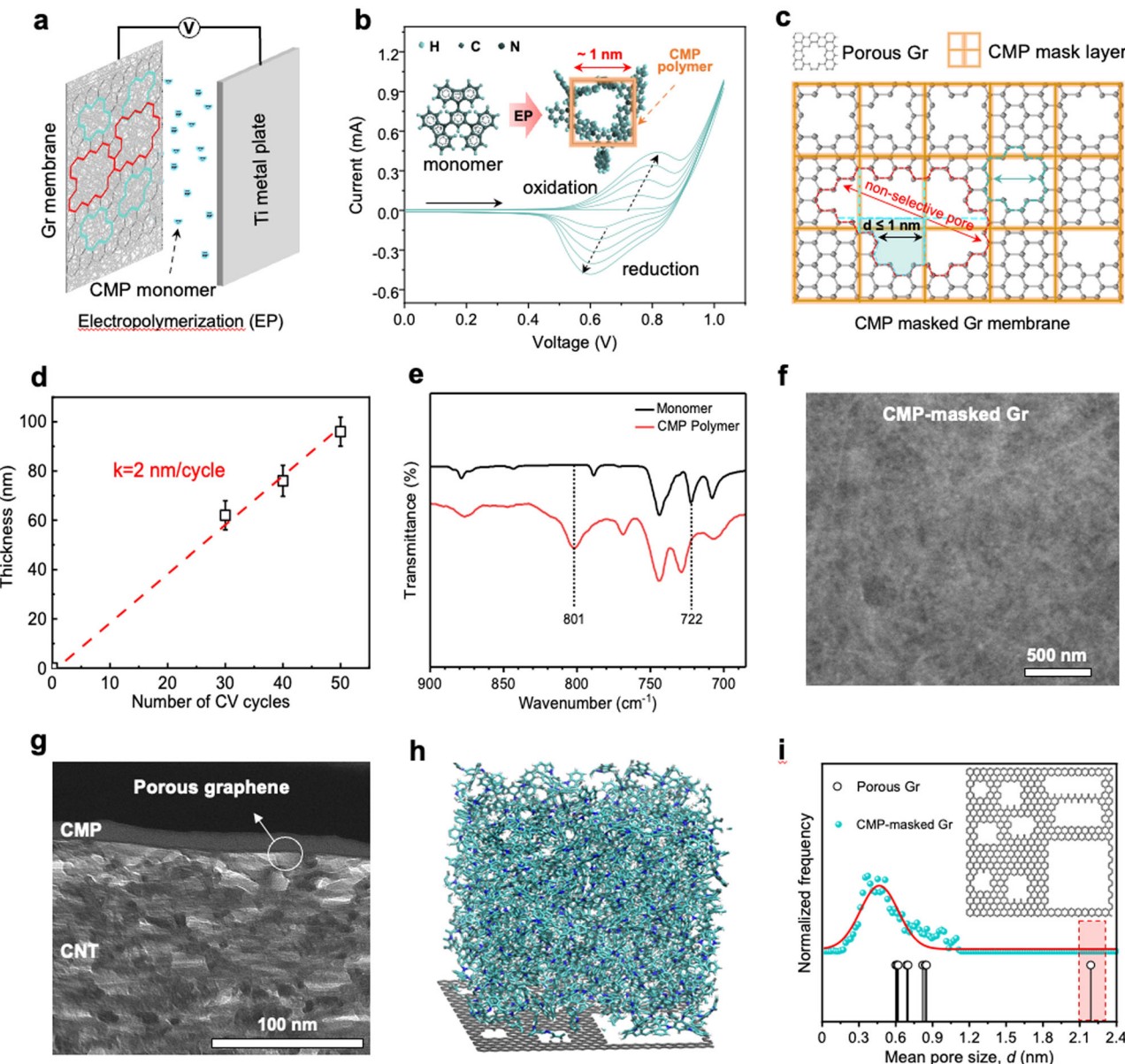

**Fig. 3 | Electrochemical repair of pore size distribution in graphene. a** Schematic illustration of the electrochemical repair device. **b** Cyclic voltammetry profiles of the electrochemical repair process recorded over 5 scan cycles. The insets show the structures of the CMP monomer and the polymer. **c** Schematic illustration of an ideal structural model of the CMP-masked Gr membrane. The red dotted area marks a non-selective large pore, and the green area shows a CMP-masked pore. **d** Membrane thickness as a function of the CV scan cycle. The error bar is the standard deviation from at least three samples, and the center of each error bar represents the average data from these samples. **e** FTIR spectra of the CMP monomer and CMP polymeric film on porous graphene. **f** Surface SEM image and **g** cross-section TEM image of the CMP-masked Gr membrane. **h** Simulated structure of the CMP-masked Gr membrane. **i** Simulated PSD of porous graphene with (top) and without CMP mask (bottom). of the CMP-masked Gr membrane. The inset illustrates the simulated structure of the porous graphene, highlighting nine representative pores of different sizes. The sizes of these pores are indicated by the peaks in the graph.

water permeance in the range of 1 to 7 bar, which indicates excellent mechanical strength and stability. Moreover, when Congo red dye was introduced into the feed, near 100% rejection was obtained (Fig. 1f), providing additional confirmation regarding the effective transfer and good structural integrity of the graphene supported by the CNT.

**Porosity incorporation in graphene for ion-ion separation**

To achieve selective ion transport from graphene, we then incorporated porosity in the graphene lattice by controlled oxidation (Fig. 2a). Considering that the hydration diameters ($D_H$) of the extensively studied monovalent salt ions ($K^+$, $Na^+$, and $Li^+$) fall within the range of 6.6–7.6 Å[40–42], and for common divalent ions ($Ca^{2+}$ and $Mg^{2+}$) within the range of 8.2–8.6 Å[43,44], pores in graphene with dimensions exceeding

3.5 Å (i.e., surpassing the steric exclusion limit), yet remaining below 8.5 Å, have the potential to yield selectivity between monovalent/divalent ions. To generate pores in graphene lattice in the size range of 3.5 to 8.5 Å, we first oxidized the graphene with $O_3$ using protocols for the generation of pores for sieving small gas molecules such as $H_2$ and $CO_2$ (kinetic diameter near 3 Å)[45,46] and later pursued controlled pore expansion. The epoxy group from oxidation with $O_3$ can be observed by X-ray photoelectron spectroscopy (XPS) (Supplementary Fig. 17b). An obvious $D$ peak in the Raman spectrum could be also observed (Supplementary Fig. 18b). However, given that we followed a protocol for the generation of small pores (~3 Å) by $O_3$, $Li^+/Mg^{2+}$ selectivity was not observed (Fig. 2e, $O_3$-Gr). This is because, both lithium ion and the larger magnesium ion encounter strong hindrances in traversing small

pores, resulting in low ion flux and poor ion selectivity. Therefore, we tried to controllably expand these pores in a $CO_2$ environment[47] to allow the transport of ions (Fig. 2a).

$CO_2$ has an exceptionally high energy barrier for pore nucleation (-5 eV)[48], while the barrier for pore expansion (-2.7 eV) can be surpassed at high temperature[47]. This distinctive property makes $CO_2$ attractive for pore expansion while avoiding new pore nucleation events. Exposure of $O_3$-treated graphene to a $CO_2$ environment at 800 °C for 5 min (referred to as "5minCO$_2$-Gr") led to pores with a high pore density of -$2.2 \times 10^{12}$ cm$^{-2}$ (Fig. 2c, d). Additional information on the pore size quantification is provided in Supplementary Note 5 (Supplementary Fig. 19). The high-temperature reaction removed most of the O-functional groups, partly by desorption and partly because these groups surround the pore and consequently are eliminated during expansion (Supplementary Fig. 17c)[45,49]. Raman spectroscopy did not show a noticeable increase in $D$ peak intensity after $CO_2$ pore expansion (Supplementary Fig. 18c).

The ion-ion separation performance of the 5minCO$_2$-Gr demonstrated a marked enhancement in the permeation rate of Li$^+$ ions. Enhanced Li$^+$ permeance (-$10^{-4}$ m/h) as well as Li$^+$/Mg$^{2+}$ selectivity (average 28.1) could be observed (Fig. 2e). Further pore expansion (e.g., 8 min treatment, referred to as "8minCO$_2$-Gr") did increase the permeation rate of Li$^+$ ions, however, Li$^+$/Mg$^{2+}$ selectivity was lost (Fig. 2e) from excessive enlargement of pores. Therefore, a 5-minute $CO_2$ treatment was selected as the optimal method. Unless explicitly stated otherwise, the designation "porous graphene" in the subsequent text pertains to the 5minCO$_2$-Gr. A comprehensive assessment of the porous graphene was conducted using ions with different sizes (Fig. 2f). The ion permeation rate followed the sequence of K$^+$ > Na$^+$ > Li$^+$ > Ca$^{2+}$ > Mg$^{2+}$, aligning with the trend in $D_H$. The K$^+$ permeation rate was close to three orders of magnitude higher compared to that from the pristine-Gr, thanks to the incorporation of ion-permeable pores by oxidation. Given that the permeation rate was one order of magnitude smaller than that of the CNT support, graphene pores governed ion transport.

## Masking of graphene pores

The potential of porous graphene in ion-ion separation can be improved if one can control the contribution of the non-selective pores in the PSD. Despite the development of many chemical and physical etching techniques[21,50–52], it has become clear that avoiding larger pores at the tail side of the PSD will be challenging. PSD for the porous graphene, collected from images of -200 pores, shows the presence of pores larger than 8.5 Å (Fig. 2d). These larger pores are usually elongated, originating from the coalescence of nearby pores during pore expansion (Fig. 2c). To improve ion-ion selectivity, the nonselective transport pathway must be blocked. For this, we report a facile electrochemical repair strategy that masks graphene pores with a CMP net. The selection of CMP as the masking net is based on several considerations. CMP has a highly interconnected and rigid microporous structure which yields a high ion flux through[53–55]. Notably, the pore size in CMP is uniform which makes it a highly predictive masking layer[56–59]. The pore size of CMP used in this study is larger than the $D_H$ of Mg$^{2+}$ ions[53]. Indeed, standalone CMP film did not yield Li$^+$/Mg$^{2+}$ selectivity (Supplementary Fig. 35). The aromatic building block of CMP is expected to have a strong interaction with the graphene by π-π interaction[55]. The interlayer spacing between graphene and CMP also does not lead to selective Li$^+$ transport, exemplified by a control experiment involving the deposition of CMP film on graphene (treated by 6-second plasma) hosting pores larger than 2 nm (Supplementary Fig. 37). These distinct characteristics enable CMP an effective masking net for large non-selective pores in graphene.

The synthesis of the CMP mask layer is carried out by a simple electrochemical route (Fig. 3a and Supplementary Fig. 21)[53], requiring a mere 2−3 minutes at room temperature. The CMP monomers used in

this study feature a spiro center. Upon crosslinking, monomers create a microporous three-dimensional (3D) conjugated network with a nanometer-sized void within the structure (Fig. 3b). In an applied electric field, the monomer undergoes a sequence of oxidation and reduction, leading to crosslinking into a uniform film over porous graphene (Fig. 3c). Additional information regarding the electropolymerization of the CMP mask layer can be found in Supplementary Note 6, Supplementary Figs. 22 and 23. The film thickness can be controlled by the reaction time (Supplementary Fig. 24). Thickness exhibits a linear correlation with the number of cyclic voltammetry (CV) scans, with a growth rate of -2 nm per CV cycle (Fig. 3d). This enables a precise and straightforward customization of the resultant CMP mask layer thickness at the nanoscale. To maintain a thin CMP layer, we opted for 5 cyclic voltammetry cycles with a scan rate of 200 mV/s, resulting in an approximate thickness of -10 nm. The resulting product from this process is referred to as "CMP-masked Gr" unless otherwise specified. Fourier-transformed infrared (FTIR) spectroscopy (Fig. 3e) revealed a peak at 801 cm$^{-1}$ [56,58] in the CMP film, which suggests the existence of carbazole crosslinking. The presence of CMP mesh on graphene was confirmed by visualizing the morphology by SEM (Fig. 3f and Supplementary Fig. 25, further details in Supplementary Note 7) and atomic force microscopy (AFM, Supplementary Fig. 26). The TEM-FIB image depicts a well-defined cross-sectional cut, with individual layers in the stack clearly resolved and identified, aligning with our expectations. A uniform CMP layer is observed with a thickness of 14 ± 3 nm, consistent with the thickness measured by AFM. Although the single atomic layer thickness of the porous graphene layer is not discernible, the clear boundary between the CMP layer and the CNT support suggests the presence of a graphene barrier. No intrusion of CNT bundles into the top CMP layer is observed. Moreover, the interface appears sharp and continuous, devoid of apparent gaps or voids, indicating a secure attachment between the CMP layer, the porous graphene, and the CNT layer. The thickness of the CMP layer increases linearly with the number of cyclic voltammetry (CV) scans, in line with the AFM results. The interfaces between these CMP layers prepared under different conditions and the porous graphene layer show no significant difference (Supplementary Fig. 27). The bottom CNT layer displays numerous voids and some cross-sectional structures of CNT bundles. The thickness of the CNT support measures 261 ± 11 nm, consistent well with the measurements obtained through SEM. The multilayer structure was also validated by the XPS depth-profiling analysis (See Supplementary Note 8 and Supplementary Fig. 28).

To corroborate the viability of the CMP masking net, we carried out molecular dynamics (MD) simulations to compute the effective PSD of a model porous graphene covered with a 6-nm-thick CMP net. Porous graphene in the simulation hosts selective and non-selective large pores (Fig. 3h and Supplementary Fig. 32). The simulation indicates that the effective size of the graphene pore shrinks due to the overlap of the pores in CMP with those in graphene. The large pores are eradicated, resulting in a narrow Gaussian distribution. This demonstrates that the CMP net can effectively divide non-selective large pores into discerning smaller ones narrowing the PSD in graphene akin to the pore-in-pore strategy used in covalent-organic frameworks (COFs) films[60] where stacking leads to overall smaller pores. Removal of larger pores should inhibit the transport of the ions with a large $D_H$, thus enhancing the ion-ion separation performance. Simulations were also conducted to investigate the interaction between CMP and the porous graphene layer. Additional details and insights can be found in Supplementary Note 12.

## Ion-ion separation from masked graphene membrane

Ion diffusion experiments using CMP net-masked graphene corroborated the impact of the electrochemical repair strategy. Mg$^{2+}$ permeation rate decreased significantly, approaching that from pristine-

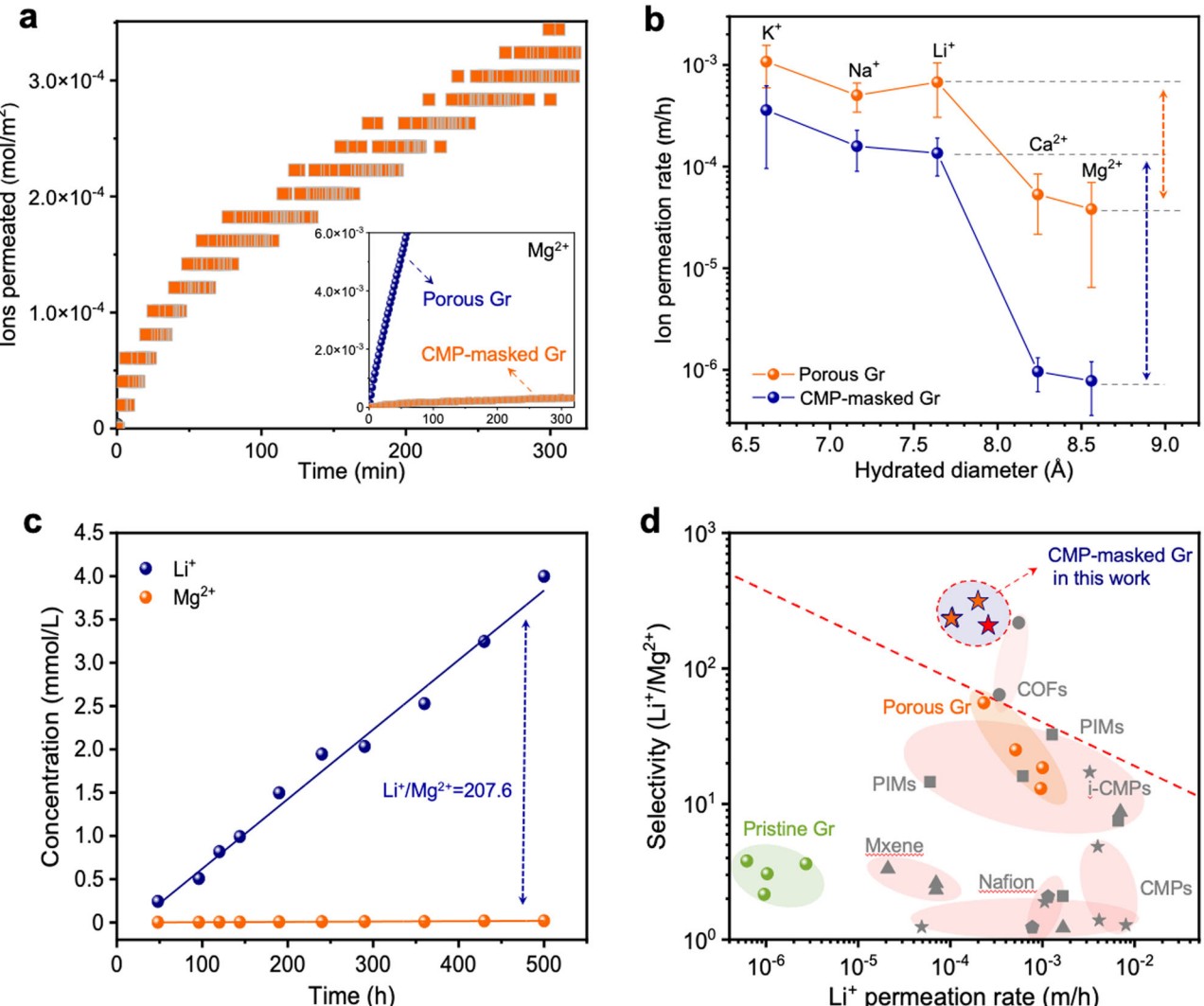

**Fig. 4 | Ion-selective transport from CMP-masked porous graphene membranes. a** Diffusion of $Mg^{2+}$ through CMP-masked Gr membranes, and the inset shows the diffusion comparison of $Mg^{2+}$ through the porous graphene membranes with and without the CMP mask layer. **b** Ion-ion separation performance of the membranes tested by different ions. The error bar is the standard deviation from at least three samples, and the center of each error bar represents the average data from these samples. **c** Long-term binary ion-sieving performance of the CMP-

masked porous graphene membrane. **d** Trade-off relationship between $Li^+/Mg^{2+}$ selectivity and $Li^+$ permeation rate of graphene membranes and state-of-the-art ion-sieving membranes reported in the literature, under concentration-driven single-ion process. The red pentagram marks the data obtained in the concentration-driven binary ion-sieving test. The red line is added to highlight the trade-off between ion permeation rate and ion-ion selectivity.

Gr (Figs. 4a, b, and 2f), confirming the blockage of its permeation from pores. Its slow permeation was indicated by a staircase pattern in its permeation over time in sharp contrast to the rapid transport trend for the $Li^+$ ion, leading to a large $Li^+/Mg^{2+}$ selectivity reaching 300, approximately 10-fold higher than that achieved from graphene pores without CMP masking (Fig. 4b). $Li^+$ permeation rate was fast $(1.3 \pm 0.6 \times 10^{-4}$ m/h), within the same order of magnitude as that of when CMP layer was not used $(6.8 \pm 3.7 \times 10^{-4}$ m/h, Supplementary Fig. 34). The high transport rate of $Li^+$ ions is due to the low resistance of CMP benefitting from the presence of a 3D coordinated net. As mentioned before, neither the standalone CMP net nor the interlayer gap between the net and graphene led to selectivity (Supplementary Figs. 35 and 38), consistent with the literature on CMP net[53], confirming that the selectivity originated from the masked graphene pores. Additional results about the optimization routine of the CMP layer and the impacts of CMP thickness on the ion-sieving performance are presented and discussed in the supporting information (Supplementary Notes 13 and 14, Supplementary Figs. 35, 37, and 38).

The ion-sieving performance tested under different pH values (4, 6.6, and 8) revealed that ion permeation rate and ion selectivity did not change significantly as a function of pH (Supplementary Fig. 39). Although zeta potential experiments showed negative charges of the CMP-masked membranes, it is likely from the strong influence of the negatively charged CNT support as widely reported in the literature[61,62] (Supplementary Note 11, Supplementary Fig. 31). We note that the pore size of the CNT support is excessively large (≥10 nm), and hence Donnan exclusion can be ruled out from the CNT support. In an extended $Li^+/Mg^{2+}$ binary ion mixture separation experiment spanning 3 weeks, stable ion flux and selectivity were observed (Fig. 4c and Supplementary Fig. 41), underscoring the robustness of the graphene membranes. Additionally, $K^+/Mg^{2+}$ and $Na^+/Ca^{2+}$ binary systems were also tested, and the CMP-masked Gr membranes showed stable selectivity, as shown in Supplementary Fig. 42. Moreover, the membranes also exhibited high performance under electric field-driven ion diffusion tests, suggesting great potential in electric field-driven applications (Supplementary Note 16, and Supplementary Figs. 44, and 45).

In summary, the ion-sieving performance of CMP-masked Gr surpasses that of most reported state-of-the-art ion-sieving membranes (Fig. 4d and Supplementary Table 2) such as COFs[63,64], polymers with intrinsic microporosity (PIM)[65], graphene oxide (GO)[66–68], transition metal carbides or nitrides (MXene)[69–71], and commercially available Nafion membranes[54,65]. Even the porous graphene lacking a CMP mask layer demonstrated comparable performance to the top-performing polymeric membrane[65], as shown in Fig. 4d. This observation underscores the substantial potential of graphene membranes for precise ion separations. Furthermore, the $K^+/Mg^{2+}$ selectivity of the best CMP-masked Gr membrane reached an impressive value of 530.4 (Supplementary Fig. 43 and Supplementary Table 3), also notably exceeding the majority of previously reported results[44,54,63,65,68,69,71,72].

Overall, we demonstrate large-area ion-sieving graphene membranes with finely tailored Å-scale pores as a promising platform for ion-ion separation. By developing the electrochemical repair strategy, non-selective pores in graphene could be effectively masked, narrowing the PSD. Large monovalent/divalent ion selectivity combined with a high monovalent ion flux, surpassing the performance of reported materials could be achieved. This selectivity is significant because, to date, studies on large-area monolayer graphene have not led to high ion-ion selectivity. This approach presents a highly promising avenue for graphene defect engineering. Moreover, this strategy facilitates the preparation of monolayer graphene membranes with customizable pore sizes, providing a promising platform for advancing diverse membrane applications.

## Methods

### Synthesis of pristine graphene
Pristine monolayer graphene used in this work was synthesized via a chemical vapor deposition (CVD) method as reported before, starting with a pre-annealed copper foil (50-μm-thick, 99.9% purity, Strem), as described in our previous work[45]. Initially, the annealed copper foil was exposed to a $CO_2$ and $H_2$ atmosphere at 1000 °C for 30 minutes, respectively, followed by exposure to a 24 standard cubic centimeter per minute (sccm) of $CH_4$ and 8 sccm of $H_2$ atmosphere with a total pressure of 460 mTorr for an additional 30 minutes. After completion of the CVD process, the reactor was cooled down to room temperature to obtain pristine single-layer graphene on the copper foil. Each batch of graphene was characterized using Raman spectroscopy before further use.

### Pore generation on monolayer graphene
To precisely introduce and tune pores into the graphene lattice, a custom-made electrical heating reactor chamber was used with an inserted thermocouple at the heating zone for precise temperature control. Briefly, the as-synthesized pristine graphene on Cu was placed inside the reactor chamber, and the system was subjected to a continuous flow of Ar gas at a rate of 100 cm and a pressure of 1 bar. The reactor temperature was maintained at 43 °C to establish thermal equilibrium. Next, the flow of Ar gas was discontinued, and an $O_3/O_2$ mixture (9% $O_3$ on a molar basis, Atlas 30, Absolute Ozone) was injected into the chamber for 1 hour. After the ozone treatment, the heating was turned off, and the reactor was purged with Ar gas to remove any remaining ozone and the reactor chamber was cooled down to room temperature. After that, the as-obtained product was then placed in a CVD furnace, and the system was purged with $H_2$ flow at a rate of 100 sccm to remove any residual air. The temperature of the furnace was then raised to 800 °C to prepare the sample for the pore generation reaction. The product obtained after this process was named $O_3$-Gr ($O_3$-treated graphene). Subsequently, 61 sccm of $CO_2$ was introduced into the furnace reactor under 800 °C. Meanwhile, the flow rate of $H_2$ was tuned to 21.3 sccm. The $CO_2$ reaction time was controlled for 5 minutes or 8 minutes for the pore expansion reaction.

The resulting product after this process was named 5min$CO_2$-Gr or 8min$CO_2$-Gr as per the treatment time.

### Fabrication of graphene membranes
To enhance the mechanical stability of the membranes, the as-obtained graphene sample on copper foil was reinforced by a CNT network film (See Supplementary Notes 1 and 2). First, the CNT film was prepared by vacuum filtrating 20 mL CNT water solution (Supplementary Fig. 1) on a commercial hydrophilic polyethersulfone (PES) substrate (0.22 μm in pore size, 50 mm in diameter, Sartorius Stedim Biotech) and then thoroughly etching the PES sacrificial layer in a dimethylformamide solution[33]. Next, the synthesized free-standing CNT film was transferred onto a graphene-Cu surface and then annealed in an Ar atmosphere at 70 °C for 1 hour to obtain CNT/Gr/Cu. Subsequently, the copper foil was etched in a $FeCl_3$ (1 mol/L) solution, leaving only the free-standing CNT-reinforced graphene membrane. To remove any remaining copper residues and other impurities, the membrane was then floated in a hydrochloric acid solution and then thoroughly rinsed in deionized (DI) water. The obtained CNT-reinforced graphene membranes were named as follows: pristine-Gr, $H_2$-Gr, 5min$CO_2$-Gr, or 8min$CO_2$-Gr, based on the type of graphene used. Finally, a PTFE porous substrate (0.1 μm in pore size, 50 mm in diameter, Nantong Longjin) was used to spoon up and support the CNT-reinforced graphene membrane for further use.

### Fabrication of CMP-masked graphene membranes
Initially, the porous graphene of 5min$CO_2$-Gr, supported by CNT, was transferred onto a PTFE porous substrate. Subsequently, a CMP mask layer was prepared on it through a scalable electropolymerization process[53]. Specifically, 28.75 mg of 1,3,5-tris(N-carbazolyl) benzene (TCB, Sigma-Aldrich) and 3.875 g of tetrabutylammonium hexafluorophosphate (TBAPF6, Sigma-Aldrich) were dissolved in a 100 mL mixture of anhydrous $CH_2Cl_2$ and $CH_3CN$ in a 1:1 volume ratio (also see Supplementary Note 6). Next, the prepared solution was loaded into a standard three-electrode electrochemical cell that was connected to an electrochemical workstation (VSP-300, BioLogic). An $Ag/Ag^+$ non-aqueous electrode was used as the reference electrode, a titanium metal plate was used as the counter electrode, and a 5min$CO_2$-Gr porous graphene membrane was used as the working electrode. Cyclic voltammetry was performed within the voltage range of −0.8 V to 1.03 V. Finally, a very thin CMP polymer film was generated and deposited onto a porous graphene lattice to serve as a mask layer. The synthesized membranes were named CMP-masked Gr and soaked in a $CH_3CN$ solution to remove any unreacted monomers and electrolytes.

### Pressure-driven nanofiltration performance tests
The nanofiltration performance tests were conducted at room temperature using a homemade permeation cell. The pressure was set to 7 bar, while the permeate side remained open to the atmosphere. A graphene membrane was tightly sealed within the membrane cell as shown in Supplementary Figs. 15 and 16. During the test, the permeate solution was collected, and its weight was monitored by a digital balance. The dye concentration was measured using a UV-vis spectrophotometer (Lambda 365, PerkinElmer). Permeance $P$ (LMH/bar) was calculated by Eq. (1), whereas the dye rejection $R$ (%) was calculated by Eq. (2):

$$P = V/(A \cdot \Delta t \cdot \Delta P) \tag{1}$$

$$R = (1 - C_P/C_F) \times 100\% \tag{2}$$

where $V$ (L) is the volume of the permeated solution collected in a certain time $\Delta t$ (h) under the pressure different $\Delta P$ (bar), and $A$ is the

membrane area (m²). $C_P$ and $C_F$ are the dye concentrations in the permeate and feed solutions, respectively.

## Concentration-driven single-ion-sieving tests

The concentration-driven sieving tests were conducted using a diffusion cell with two chambers (Supplementary Fig. 20). A membrane was securely placed at the joint between the two chambers. Each of the chambers was filled with 50 mL of deionized water and 50 mL of salt solution (0.1 mol/L). Magnetic stirring was used in both chambers to ensure mixing. The ion permeation rate was measured automatically by an ICP-OES-calibrated conductivity meter (Mettler-Toledo GmbH, SevenCompact Cond. Meter S230). The ion permeation rate (flux) normalized by the concentration difference (mol/m³), $J$ (m/h), was calculated using Eq. (3), and selectivity, $S$, was calculated based on Eq. (4):

$$J = C \cdot V/(A \cdot \Delta t)/C_\Delta \tag{3}$$

$$S = \frac{J_a/C_a}{J_b/C_b} \tag{4}$$

where $C$ (mol/L) and $V$ (L) are the concentration and the volume of the permeate solution, respectively, $A$ is the membrane area (m²), $\Delta t$ (h) is the test time, and $C_\Delta$ is the concentration difference between the two chambers. $J_a$ (m/h) and $J_b$ (m/h) are the ion permeation rate of ion $a$ and ion $b$, respectively, and $C_a$ (mol/L) and $C_b$ (mol/L) are the concentration of the feed solution of ion a and ion b, respectively.

## Concentration-driven binary ion-sieving tests

50 ml of mixture salts solution containing 0.1 mol/L A salt and 0.1 mol/L B salt was used as the feed solution, and the permeate chamber was filled with 50 ml of deionized water. The concentrations of ions on the permeate side were measured by inductively coupled plasma-optical emission spectrometry (ICP-OES, 5110, Agilent).

## Electric field-driven ion-sieving tests

The electric field-driven ion-sieving tests were conducted using an electrochemical workstation (VSP-300, BioLogic) with an electrochemical cell (Supplementary Fig. 44). The membrane was placed between the two chambers of the cell, and each chamber was filled with a 10 mmol/L salt solution. Additionally, a pair of Ag/AgCl electrodes were carefully positioned in the chambers. Ionic conductance was determined within the voltage range of −0.2 to 0.2 V, with a step size of 0.02 V/s, and the resulting current was recorded as a function of the applied voltage. The selectivity, $S_e$, is defined based on Eq. (5),

$$S_e = \frac{G_a}{G_b} = \frac{I_a/U_a}{I_b/U_b} \tag{5}$$

where $G$ (S), $I$ (A), and $U$ (V) represent average electrical conductance, average current, and potential, respectively, and subscripts a and b represent ion a and b, respectively.

## Construction of membrane models

The structure of CMP was obtained from the literature[53]. Initially, the simulation system underwent thorough relaxation via constant-temperature, constant-pressure (NPT) simulations at 300 K and 1.0 atm. Subsequently, to further relax the membrane, a 10 ns molecular dynamics simulation was executed at 300 K following the 40-cycle annealing process, resulting in a stable membrane structure. This established structure was used to generate a histogram of N···N distances between two adjacent TCB moieties. The pore size was determined based on the probability of N···N distances indicated by the histogram. The free volume of the constructed membrane system was

simulated using atomic volume and surface. PSD was analyzed using Zeo++[73]. The above calculations were performed using GROMACS 4.6.7, GROMOS force field, and PRODRG modules[74–76].

## Characterizations

Pristine single-layer graphene and O₃-treated graphene were imaged using an aberration-corrected high-resolution transmission electron microscope (FEI Titan Themis) equipped with a Wein-type monochromator at the operating voltage of 80 kV to mitigate the knock-on damage, the negative spherical aberration (Cs) of -18 μm was applied to enhance the resolution of imaging. For sample preparation, we followed the method described in previous work[45]. In brief, the graphene-coated with polymer was transferred onto a silicon nitride grid with an array of 1-μm-hole. Before any further treatments or imaging processes, the polymer coating was removed by washing the grid with heptane at least three times. CNT-graphene membranes were directly transferred to 400 mesh carbon film-coated TEM grids. TEM images and SAED images of CNT-graphene membranes were collected using FEI Tecnai G2 Spirit Twin at an operating voltage of 120 kV and FEI Talos F200s at an operating voltage of 80 kV. Focused ion beam (FIB) was conducted to check the cross-section of the samples. To achieve TEM cross-section images with enhanced contrast, the porous Gr-CNT membranes were transferred onto a silicon wafer coated with a 100-nm thick Au metal layer. Following the electrochemical repair process under varying conditions, the CMP-Gr-CNT-Au-Si samples were once again coated with a 100-nm thick layer of Au. Successively, from top to bottom, the TEM-FIB samples consist of six layers: An Au metal layer, a CMP mask layer, a porous graphene layer, a CNT support layer, a second Au metal layer, and the silicon wafer. To prepare the TEM lamella, a 1μm-thick protective amorphous carbon film was deposited onto the sample using a 150 pA ion beam at 30 kV. Subsequently, a coarse milling and thinning procedure was executed, involving 6.5 nA for coarse milling and 3 nA/1.5 nA/0.7 nA for thinning. A final polishing step at low voltage (5 kV) and 30 pA for 10 seconds was conducted to eliminate carbon contamination generated during the FIB lamella preparation process. FIB lamella sample was imaged using FEI Talos F200s at an operating voltage of 80 kV. FEI Teneo SEM was used to obtain SEM images at 1.0−5.0 kV and working distances of 3−8 mm. Raman measurements (Renishaw inVia) were performed on graphene-Cu foil immediately after the synthesis and pore etching using a 457 nm excitation laser. XPS measurements were conducted on graphene-Cu foil using an Axis Supra instrument (Kratos Analytical). The pass energy was 20 eV, and the step size was adjusted to 0.1 eV. Peak fitting was carried out using the software CasaXPS, and for background subtraction, the Shirley method was employed. The surface roughness and thickness of the samples were analyzed by atomic force microscopy (AFM, MultiMode, Bruker). The chemical structure was characterized by Fourier-transform infrared spectroscopy (FTIR, Spectrum Two, PerkinElmer). Zeta potential values were determined by an Anton Paar solid surface analyzer (SurPASS 3, Anton Paar). XPS etching experiments were conducted through XPS (Kratos AXIS165) equipped with Al Kα radiation.

## Data availability

The authors declare that the data supporting the findings of this study are available within the paper and its supplementary information files. The source data for the figures in the main text can be obtained from https://doi.org/10.5281/zenodo.10912438. The source data is provided as a source data file. Source data are provided with this paper.

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

## Acknowledgements

We acknowledge our home institution, EPFL, for supporting the work via the EPFLeaders4Impact program. This project has received funding from the European Union's Horizon 2020 research and innovation program under the Marie Skłodowska Curie grant agreement No. 101034260. We also acknowledge GAZNAT and Prof. Jun Ma for help with the project.

## Author contributions

Z.Z. conceived and designed the experiments under the guidance of K.V.A. The original draft was prepared by Z.Z. and reviewed, validated, and edited by K.V.A. Z.Z., and K.Z. prepared the membranes, and Z.Z. contributed to most of the experiment and data analysis. K.Z. performed ICP-OES and helped in CVD and ion-sieving tests. H-Y.C. carried out AC-HRTEM sample preparation and imaging. Y.S. studied the cross-section of the film. S.S. conducted FTIR and AFM. K.H. performed XPS. M.C. performed Raman. W.S. performed simulations. All authors discussed the results and commented on the manuscript.

## Competing interests

The authors declare no competing interests.
