## [Peer Review File · Nature Communications]

Electrochemical-repaired porous graphene membranes for precise ion-ion separationREVIEWER COMMENTS

Reviewer #1 (Remarks to the Author):

This manuscript describes a new method to repair and narrow down non-selective pores in single-layer graphene membranes for ionic separations. The membranes exhibit an extremely high selectivity between lithium and magnesium, which is important for lithium extraction from brines and other applications. The membranes consist of a porous graphene on a carbon nanotube (CNT) mat, followed by electropolymerization of a conjugated microporous polymer (CMP) on the graphene layer. All steps in the membrane fabrication can potentially be scaled up. Pressure-driven and diffusion-driven transport measurements are reported, along with mixed salt separations for lithium and magnesium. The work is innovative and thorough with detailed characterization and well-designed controls. It is a significant advance for nanoporous graphene membranes and is likely to be of broad interest due to the importance of lithium separations for a sustainable economy. The reviewer recommends publication after addressing the minor points below.

Figure 2d – It will be helpful to state the approximate number density of pores in addition to the relative pore size distribution shown in the figure.

Page 9 – It is not clear in the main text, methods, or supplementary that the electropolymerization was carried out on the graphene after transfer to CNT and PTFE. It should be stated clearly.

In the control experiment to show that the selectivity arose from the graphene pores and not from the CMP, it seems that the control experiment to polymerize CMP on CNTs did not fill in the gaps to create a continuous layer of CMP. If that is the case, it is not possible to conclude that the CMP did not contribute to selectivity, and a control experiment where the CMP entirely fills the gaps between the CNTs is needed. The control experiment with graphene containing larger pores more strongly supports the conclusion that the selectivity arises from the graphene pores.

It is stated in the SI page 8 that only sintering of polymers has been reported as a masking layer. However, polyelectrolyte coating has been reported in the past to mask defects in nanoporous graphene (<https://doi.org/10.1002/adma.202108940>). Other methods such as polymerization at defect sites have also been reported (e.g., <https://doi.org/10.1021/acs.nanolett.0c01934>). It would be relevant to provide this context in the main text.

Figure 4d – The comparison makes sense only when the concentration difference driving the transport of ions is the same. E.g., a membrane that has been tested, say, at 1 mM concentration difference would appear to have 3 orders of magnitude lower performance on the x-axis than a membrane tested at 1 M concentration difference. It would be better to normalize by the concentration difference and report the x-axis in units of $(\text{mol}/\text{m}^2\text{-h})/(\text{mol}/\text{m}^3) = \text{m}/\text{h}$ or m/s .

There are many SEM images of graphene and CNT, but relatively few of the CMP layer. It will be helpful to provide some more images showing the cross-section of the CMP similar to Fig. 1b.

The simulations of the CMP are not adequately described. It is not clear what exactly was simulated. It seems that interaction between CMP and graphene was simulated rather than polymerization. It is also not clear how the pore size distributions were obtained, and whether the pores were stable or not. How the CMP interacts with the pores is also not clear. Addition of a much more detailed supplementary note is warranted. If possible, a simulation of the control with larger pores (6s plasma) will be helpful in providing insights as to how CMP works.

Minor typos:

SI note S1 – should be polydopamine instead of dopamine?

Page 2, line 48 – The sentence “For example, a masking layer composed of a Å-scale aperture would reduce the selectivity-deteriorating rapid nonselective transport from nanoscale tears and

cracks as well as nanometer-sized large nonselective pores in graphene" could be rephrased for better clarity.

Page 3, line 64 – "resist the thickness" should be "limit the thickness"

Page 7, line 157 – "an noticeable" to "a noticeable"

Page 10, line 235 – Consider rephrasing "staircase pattern" – presumably this is caused by digitization of the measurement?

SI, page 6 – "the detector is used"

Fig. S2 – The caption does not make it clear what is depicted in the SEM images.

Reviewer #2 (Remarks to the Author):

This manuscript, authored by Chen et al., presents a comprehensive investigation into the melamine-modulated interfacial polymerization. The incorporation of melamine results in the narrowing of pore size in the polyamide membrane through intermolecular hydrogen bonds. The study includes multicomponent salt rejection tests, revealing the significant influence of anions on the rejection of divalent ions. Overall, the manuscript is well-written. However, the main concern is the lack of novelty, given the extensive prior research on intermolecular hydrogen bond interactions in polyamide membranes. As such, this manuscript may not be suitable for publication in this high-impact journal. Below list some major comments for improvement.

1: There is not sufficient evidence to support the melamine-induced in-plane tendency in structure. The author concludes that the overall planarity of the structure is induced by melamine through the comparison of the planarity of cyclic structures formed by monomer connections. However, the true structures of polyamide membranes are complicated. The cyclic structure is only a small part of its overall structure, and larger-scale molecular simulations are needed to support the author's conclusion.

2: In the membrane performance tests, it appears that a clear explanation has not been established regarding the relationship between the suppression of structural unit distortion and the variations in pore size, membrane thickness, and water flux when melamine is added.

Reviewer #3 (Remarks to the Author):

This paper reports the fabrication of CMP-masked graphene membranes for ion separation. The methodology of creating and healing the graphene pores is interesting, with outperforming Li⁺/Mg²⁺ selectivity. However, the authors need to demonstrate a better understanding of the interaction between the CMP-masked layer and graphene pores. Moreover, the current ion sieving test through self-diffusion process is less applicable. This paper may be published in Nature Communications if the comments below are addressed.

1. The authors attributed the change of pore size to the stacking of CMP and graphene pores, but the key question is whether the CMP layer was only deposited on the graphene surface or intruded into the pores. To address this concern, the authors could carry out an XPS etching experiment to explore the in-depth profile of the CMP layer.

2. What is the surface charge of the membranes? Fig.S20 shows positive charged amines in the CMP backbones, which could significantly contribute to the high Li⁺/Mg²⁺ selectivity as proved by many reports in literature. This may not be pronounced for CMP films alone since their pore size was too loose to create Donnan effect, as shown in Table S2. However, if the CMP was intruded into graphene pores, the effect of charge could be more important than that of the pore size for achieving the high Li⁺/Mg²⁺ selectivity.

3. In Table S2, the ion concentration used in this paper was 0.1M, whereas most the comparative work used higher ion concentrations. This is not a fair comparison as concentration polarization

could be more severe at the membrane surface for higher ion concentrations.

4. The pristine graphene membranes in Fig. 1e showed good stability in pressure driven filtration up to 8 bar, why was the ion sieving experiment through CMP-masked graphene membranes not performed under the same conditions instead of self-diffusion? Self-diffusion is a less attractive process for the real industrial application. Also, the high Li⁺/Mg²⁺ selectivity in self-diffusion process may not reflect the actual selectivity in the real operation under high pressures.

Reviewer #4 (Remarks to the Author):

The work entitled Electrochemical-repaired porous graphene membranes for precise ion-ion separation by Zhou et al. presents experimental and theoretical investigations on the ion separation by means of nanoporous graphene membranes. I believe that the covered topic is interesting for a broad community working in advanced filtration field.

The manuscript details a meticulous study of nanoporous graphene for ion-ion separation and the use conjugated microporous polymer to statistically reduce the pore size of nonselective larger pores. The nanoporous graphenes used in the current work are the same ones that were used in previous works (for instance Huang et al., *Adv. Mater.* 2022, 2206627 or Lee et al., *ACS Nano.* 16, 15382–15396, 2022). One half of the paper is reporting these previous results. This does raise questions regarding the novelty of the current manuscript. What mainly distinguishes this paper from previous works is that this time they used well-established support membrane to improve ion selectivity.

The work is very comprehensive, well carried out, and the scientific presentation is good. The authors certainly deserve credit for this study, and this paper should get. However, my concern with publishing this study in *Nature Communications* is that there is no real material novelty. The work seems to supplement the previous papers by the same authors. Therefore, I am hesitant to recommend publication in *Nature Communications*.

Anyway, here some issues that could clarify and potentially improve this draft paper.

1. The authors should clarify if the pore edges are hydrogenated or not.
2. The authors should further elaborate on the pore diameter quantification. In figure 2d, the histogram displays bars of 0.02 nm. How they calculate the pore diameter? What's the associated error? Has sense to have around pores with a diameter of 0.02 nm?
3. Why pristine-Gr in figure 2f doesn't have error bars?
4. How many samples are studied in both configurations CNT-nanoporous graphene and CNT-nanoporous graphene- CMP mask? Figure 2e displays sets of 4 and 3 points, why? In figure 2e, are the four points indicated as Support four samples? Are those four samples the same ones indicated as Pristine-Gr, and then, as O3-Gr and so on? Are the CMP masked GR in figure 4d the same samples displayed in figure 2e?
5. The authors claim that CMP-mask is playing a role as pore size reducer. Then, they should further explain why figure S28 displays more than one order of magnitude in selectivity against the CMP mask thickness. One should expect a rough constant selectivity and a continuous increasing with increased thickness. A cross-section TEM image would be beneficial to further understand the CNT-nanoporous graphene – CMP mask interface for different CMP synthesis conditions.
6. I suggest to perform Raman of the CMP-masked Gr to put some light into the effect of the electropolymerization on the nanoporous graphene.

In conclusion, I am inclined to not recommend for publication in *Nature Communications* journal

RESPONSE TO REVIEWERS' COMMENTS

Reviewer 1

This manuscript describes a new method to repair and narrow down non-selective pores in single-layer graphene membranes for ionic separations. The membranes exhibit an extremely high selectivity between lithium and magnesium, which is important for lithium extraction from brines and other applications. The membranes consist of a porous graphene on a carbon nanotube (CNT) mat, followed by electropolymerization of a conjugated microporous polymer (CMP) on the graphene layer. All steps in the membrane fabrication can potentially be scaled up. Pressure-driven and diffusion-driven transport measurements are reported, along with mixed salt separations for lithium and magnesium. The work is innovative and thorough with detailed characterization and well-designed controls. It is a significant advance for nanoporous graphene membranes and is likely to be of broad interest due to the importance of lithium separations for a sustainable economy. The reviewer recommends publication after addressing the minor points below.

Author's Response: We thank the reviewer for constructive comments which helped to improve the manuscript.

Question 1:

Figure 2d – It will be helpful to state the approximate number density of pores in addition to the relative pore size distribution shown in the figure.

Author's Response: We are grateful for this helpful suggestion. In the revised manuscript, the pore density has been incorporated into the main text, and the statistical analysis method has been included in Supplementary Note S5 (as reproduced below for your convenience).

(Main text, line 158)

"Exposure of O₃-treated graphene to a CO₂ environment at 800 °C for 5 min (referred to as "5minCO₂-Gr") led to pores with a high pore density of $\sim 2.2 \times 10^{12} \text{ cm}^{-2}$. (Figures 2c-d). Additional information on the pore size quantification is provided in Supplementary Note S5 (Figure S19)."

(Revised Supporting Information)

Supplementary Note S5

Pore size quantification

The pore diameter is determined through the analysis of TEM images. After the image processing, a sphere fitting method is employed to determine the effective pore diameter. Specifically, a circle is fitted inside a pore by locating the outermost carbon atoms at the edge. The resulting diameter (D_i) is obtained as illustrated in Fig. S19. This circle contacts the outermost edge of the carbon atoms. Subsequently, adding the atomic radii of carbon atoms, we constructed another circle intersecting the center of these edge carbon atoms. The diameter of this circle is denoted as D_c . Next, we subtracted the nonbonded interaction length between graphitic carbon and a water molecule, represented by $2^{1/6} \sigma$ ($\approx 3.86 \text{ \AA}$) corresponding to the position of the potential well^[1], from D_c . This subtraction yields an effective pore diameter.

Fig. S19 Schematic of the measurement of the pore size in the graphene.

Question 2:

Page 9 – It is not clear in the main text, methods, or supplementary that the electropolymerization was carried out on the graphene after transfer to CNT and PTFE. It should be stated clearly.

Author’s Response: We apologize for missing out on clarity on this important aspect of preparation. Indeed, electropolymerization was carried out on graphene after transfer to CNT and PTFE. While it is ideal to do it on graphene resting on Cu, we observed issues from the Cu side reaction during electropolymerization.

A more detailed description of the electropolymerization processes has been included in the method part and Supplementary Note S6. A Schematic of an electrochemical-repair device has also been added as shown below. The added text is reproduced below.

Fig. S21 Schematic of the electropolymerization device.

(Method section, lines 358-372)

Fabrication of CMP-masked graphene membranes

Initially, the porous graphene of 5minCO₂-Gr, supported by CNT, was transferred onto a PTFE porous

substrate. Subsequently, a CMP mask layer was prepared on it through a scalable electropolymerization process^[2, 3]. Specifically, 28.75 mg of 1,3,5-tris(N-carbazolyl) benzene (TCB, Sigma-Aldrich) and 3.875 g of tetrabutylammonium hexafluorophosphate (TBAPF6, Sigma-Aldrich) were dissolved in a 100 mL mixture of anhydrous CH₂Cl₂ and CH₃CN in a 1:1 volume ratio (also see Supplementary Note S6). Next, the prepared solution was loaded into a standard three-electrode electrochemical cell that was connected to an electrochemical workstation (VSP-300, BioLogic). An Ag/Ag⁺ nonaqueous electrode was used as the reference electrode, a titanium metal plate was used as the counter electrode, and a 5minCO₂-Gr porous graphene membrane was used as the working electrode. Cyclic voltammetry was performed within the voltage range of -0.8 V to 1.03 V. Finally, a very thin CMP polymer film was generated and deposited onto a porous graphene lattice to serve as a mask layer. The synthesized membranes were named CMP-masked Gr and soaked in a CH₃CN solution to remove any unreacted monomers and electrolytes.

(Revised Supporting Information)

Supplementary Note S6

Electropolymerization of CMP mask layer

In this study, 1,3,5-tris(N-carbazolyl) benzene (TCB) served as the building block for CMP. Molecular simulations^[2] indicated its capability to form a CMP polymer with a pore size from 0.85 to 1 nm, surpassing the D_H of Mg²⁺ ions. Consequently, the CMP layer is not anticipated to exhibit selectivity for mono/divalent ions. Both the experimental findings in this study and that in the literature^[2] corroborate that the CMP layer demonstrates negligible Li⁺/Mg²⁺ selectivity (further discussion in Supplementary Notes S13 and S14), aligning with the simulation predictions.

The polycarbazole CMP mask layer was synthesized through continuous cyclic voltammetry (CV) in the voltage range of -0.80 V to 1.03 V (*vs.* Ag/Ag⁺) at a scanning rate of 200 mV/s. The CV curve is composed of both a positive and a negative segment. In the positive CV scan, the carbazole units of TCB underwent oxidation^[2, 3], with the onset of oxidative potential observed at around 0.9 V. The resulting carbazole radicals subsequently coupled to form dimeric carbazole cations, as illustrated in Fig. S22. A reductive peak appeared at 0.64 V in the negative scan (Fig. S23), indicative of the reduction of the dimeric carbazole cations. From the second scan onward, a new and stable oxidative peak emerged at 0.75 V, corresponding to the oxidation of the formed dimeric carbazoles^[3]. Subsequent cathodic sweep scans reduced the cation radicals back to their neutral state. Further reactions extended the polymer chain and reduced the solubility. Ultimately, the CMP film was deposited *in situ* on the porous graphene. The film's thickness could be precisely controlled by adjusting the number of CV scans^[3]. The membrane thickness increased linearly corresponding to the number of CV scans, exhibiting a growth rate of ~2 nm per CV cycle as discussed in the main text (Figs. S24 and 3d). Notably, achieving such precise control over the thickness of a polymer film is challenging using traditional synthetic methods.

Question 3:

In the control experiment to show that the selectivity arose from the graphene pores and not from the CMP, it seems that the control experiment to polymerize CMP on CNTs did not fill in the gaps to create a continuous layer of CMP. If that is the case, it is not possible to conclude that the CMP did not contribute to selectivity, and a control experiment where the CMP entirely fills the gaps between the CNTs is needed. The control experiment with graphene containing larger pores more strongly supports

the conclusion that the selectivity arises from the graphene pores.

Author's Response: As a control, we prepared CMP on the CNT support without graphene. This was done under different scan rates (50, 200, and 300 mV/s) with 5 CV cycles. The performance of these samples was compared (Fig. S35) and discussed in Supplementary Note S14.

(Revised SI, Supplementary Note S13).

“It should be noted that 5c50CMP-CNT (50 mV/s, 5 CV cycles) features a continuous CMP layer on the CNT network, fully filling the gaps between CNT bundles, as confirmed by the SEM images (Fig. S36). The membrane's surface exhibits typical granular structures, with the underlying CNT being invisible. Although the membranes 5c50CMP-CNT have a continuous structure, the $\text{Li}^+/\text{Mg}^{2+}$ selectivity is still negligible with a value of 1.2. It indicates that the porous graphene layer plays the role of the selective layer in the membranes, and the CMP layer itself is nonselective to the $\text{Li}^+/\text{Mg}^{2+}$.”

Fig. S36 SEM images of the membrane 5c50CMP (scan rates 50 mV/s, 5 CV cycles) on CNT support (without graphene) under different magnifications.

The other two CMP-CNT membranes prepared using 200 and 300 mV/s also did not exhibit any $\text{Li}^+/\text{Mg}^{2+}$ selectivity. Consequently, we concluded that the CMP layer itself does not possess $\text{Li}^+/\text{Mg}^{2+}$ selectivity.

As an additional control, we prepared graphene membranes using CMP mask where we intentionally prepared large pores in graphene. This was done by using O_2 plasma treatment of graphene with a plasma time of 6 s which is known to make large pores (2.4 nm on average)^[4]. In this case, the $\text{Li}^+/\text{Mg}^{2+}$ selectivity was lower than 3 (Fig. S37), due to the large pore size of the 6s plasma-treated graphene. It strongly supports the conclusion that the selectivity arises from CMP-masked graphene pores when pores in graphene are created by a controlled method to yield pore size distribution for ion-ion separation.

Question 4:

It is stated in the SI page 8 that only sin-coating of polymers has been reported as a masking layer. However, polyelectrolyte coating has been reported in the past to mask defects in nanoporous graphene (<https://doi.org/10.1002/adma.202108940>). Other methods such as polymerization at defect sites have also been reported (e.g., <https://doi.org/10.1021/acs.nanolett.0c01934>). It would be relevant to provide this context in the main text.

Author's Response: We express our gratitude for sharing the important information, and we have incorporated this valuable context into the main text, citing them as Ref. 30 and 31. The relative

description of spin coating has been removed in the revised SI.

Question 5:

Figure 4d – The comparison makes sense only when the concentration difference driving the transport of ions is the same. E.g., a membrane that has been tested, say, at 1 mM concentration difference would appear to have 3 orders of magnitude lower performance on the x-axis than a membrane tested at 1 M concentration difference. It would be better to normalize by the concentration difference and report the x-axis in units of $(\text{mol}/\text{m}^2\text{-h})/(\text{mol}/\text{m}^3) = \text{m}/\text{h}$ or m/s .

Author’s Response: We appreciate the suggestion. The units have been normalized as suggested and all the figures have been updated. We have reproduced the figure 4 below.

Figure 4. Ion selective transport from CMP-masked porous graphene membranes. a) Diffusion of Mg²⁺ through CMP-masked Gr membranes and the inset shows the diffusion comparison of Mg²⁺ through the porous graphene membranes with and without the CMP mask layer. b) Ion-ion separation performance of the membranes tested by different ions. c) Long-term binary ion sieving performance of the CMP-masked porous graphene membrane. d) Trade-off relationship between Li⁺/Mg²⁺ selectivity and Li⁺ permeation rate of graphene membranes and state-of-the-art ion-sieving membranes reported in the literature, under concentration-driven single-ion process. The red pentagram marks the data obtained in the concentration-driven binary ion sieving test. The red line is added to highlight the trade-off between ion permeation rate and ion-ion selectivity.

Question 6:

There are many SEM images of graphene and CNT, but relatively few of the CMP layer. It will be helpful to provide some more images showing the cross-section of the CMP similar to Fig. 1b.

Author's Response: We attempted to observe the boundary between the CMP layer and Gr-CNT layer from a perspective similar to Fig. 1b. However, while in Figure 1b we benefit from the electron transparency of graphene, we lose this advantage when CMP is deposited on graphene. As per reviewer's suggestion, we gathered cross-sectional images where cross-section was attained by focused ion beam (FIB). The results have been added to the manuscript and SI, and reproduced below. Briefly, we observe a sharp interface between CMP on graphene/CNT layer without any significant intrusion of CMP in CNT layer. This nicely shows that CMP makes a flat interface with graphene.

(Main text, characterizations section, lines 440-452)

Focused ion beam (FIB) was conducted to check the cross-section of the samples. To achieve TEM cross-section images with enhanced contrast, the porous Gr-CNT membranes were transferred onto a silicon wafer coated with a 100-nm thick Au metal layer. Following the electrochemical repair process under varying conditions, the CMP-Gr-CNT-Au-Si samples were once again coated with a 100-nm thick layer of Au. Successively, from top to bottom, the sample consists of seven layers: An Au metal layer, a CMP mask layer, a porous graphene layer, a CNT support layer, a second Au metal layer, and the silicon wafer. To prepare the TEM lamella, a 1 μ m-thick protective amorphous carbon film was deposited onto the sample using a 150-pA ion beam at 30 kV. Subsequently, a coarse milling and thinning procedure was executed, involving 6.5 nA for coarse milling and 3 nA/1.5 nA/0.7 nA for thinning. A final polishing step at low voltage (5 kV) and 30 pA for 10 seconds was conducted to eliminate carbon contamination generated during the FIB lamella preparation process. FIB lamella sample was imaged using FEI Talos F200s at an operating voltage of 80 kV.

(Lines 227-240)

The TEM-FIB image depicts a well-defined cross-sectional cut, with individual layers in the stack clearly resolved and identified, aligning with our expectations. A uniform CMP layer is observed with a thickness of 14 ± 3 nm, consistent with the thickness measured by AFM. Although the single atomic layer thickness of the porous graphene layer is not discernible, the clear boundary between the CMP layer and the CNT support suggests the presence of a graphene barrier. No intrusion of CNT bundles into the top CMP layer is observed. Moreover, the interface appears sharp and continuous, devoid of apparent gaps or voids, indicating a secure attachment between the CMP layer, the porous graphene, and the CNT layer. The thickness of the CMP layer increases linearly with the number of cyclic voltammetry (CV) scans, in line with the AFM results. The interfaces between these CMP layers prepared under different conditions and the porous graphene layer show no significant difference (Figure S27). The bottom CNT layer displays numerous voids and some cross-sectional structures of CNT bundles. The thickness of the CNT support measures 261 ± 11 nm, consistent well with the measurements obtained through SEM.

Fig. S27 a) and c) Cross-section TEM images of the CMP-masked Gr membranes, b) schematic structures of the membrane, d), e), and f) cross-section TEM images of the CMP-masked Gr membrane prepared under different CMP synthesis conditions (with 5, 10, and 15 CV cycles, at the scanning rate of 200 mV/s).

Question 7:

The simulations of the CMP are not adequately described. It is not clear what exactly was simulated. It seems that interaction between CMP and graphene was simulated rather than polymerization. It is also not clear how the pore size distributions were obtained, and whether the pores were stable or not. How the CMP interacts with the pores is also not clear. Addition of a much more detailed supplementary note is warranted. If possible, a simulation if the control with larger pores (6s plasma) will be helpful in providing insights as to how CMP works.

Author's Response: We thank the reviewer for valuable and helpful suggestions which helped in improving the manuscript. We revised the manuscript with data from additional simulations and more details on the simulation. The CMP layer was synthesized using electropolymerization where the CMP layer served as a mask for graphene pores. In the simulations, the structure of the CMP-masked Gr was investigated, and the effective pore size distribution (PSD) of the masked graphene pore was calculated. Pore size distribution was analyzed using Zeo++, which is a widely used and efficient algorithm for PSD calculation^[2, 5-8]. Briefly, a Voronoi network was first obtained by performing a Voronoi decomposition on atoms contained in the porous CMP material. The obtained Voronoi network represents the void space in the porous material^[9]. The information was used to calculate the probe's accessible surface area and accessible volume^[10]. The PSD was then analyzed based on these data. A more detailed description of the CMP simulations has been added in the revised version, and it is reproduced here.

(Methods section, starting at line 418)

Construction of membrane models

The structure of CMP was obtained from the literature^[2]. Initially, the simulation system underwent thorough relaxation via constant-temperature, constant-pressure (NPT) simulations at 300 K and 1.0 atm. Subsequently, to further relax the membrane, a 10 ns molecular dynamics simulation was executed at 300 K following the 40-cycle annealing process, resulting in a stable membrane structure. This established structure was used to generate a histogram of N···N distances between two adjacent TCB moieties. The pore size was determined based on the probability of N···N distances indicated by the histogram. The free volume of the constructed membrane system was simulated using atomic volume and surface. PSD was analyzed using Zeo++^[9]. The above calculations were performed using GROMACS 4.6.7, GROMOS force field, and PRODRG modules^[11-13].

Therefore, we believe the membrane pores in the simulations are stable after sufficient relaxation and annealing processes. The pores in the real membranes are stable too, which is supported by the 500 hours $\text{Li}^+/\text{Mg}^{2+}$ testes (Fig. S 4c) and 100 hours $\text{K}^+/\text{Mg}^{2+}$ and $\text{Na}^+/\text{Ca}^{2+}$ testes (Fig. S42).

The interaction between CMP and graphene was further investigated in the revised version. We added simulations for CMP-masked graphene where pores in graphene are large and nonselective (representing a control sample where we experiment using 6 s plasma to prepare nonselective graphene). The results have been discussed and added to the revised SI and are provided below.

(Revised Supporting Information)

Supplementary Note S12

Simulation analysis of the interaction between CMP and graphene

In the main text, simulations are utilized to explore the pore size distribution (PSD) at the interface of the CMP-masked Gr membrane. These simulations are essential because masked graphene pores cannot be visualized in TEM. Herein, the focus is on investigating the interaction between CMP and porous graphene, where pores in graphene would represent selective membrane (e.g., small pores generated by 5 CO_2 -Gr) and nonselective membrane (e.g., large pores generated by 6s plasma-Gr).

For the selective membrane, eight small (selective) and large (non-selective) pores were generated. This represented a sample obtained after a 5-minute CO_2 treatment (5min CO_2 -Gr). For the nonselective membrane, four large pores were generated with sizes close to 2.4 nm. This represented graphene treated with plasma for 6 seconds (6s plasma-Gr) where pores are approximately ~2.4 nm on average, as reported in our previous work^[4].

Next, we conducted simulations for 5min CO_2 -Gr. The atomic position in the film was frozen to keep all the positions fixed. Upon achieving system stability, the 5min CO_2 -Gr graphene was replaced by the 6s plasma-Gr graphene, keeping the CMP film the same. The structure of 5min CO_2 -Gr and 6s plasma-Gr are different, but the lattice positions were the same. Therefore, the “CMP masked 5min CO_2 -Gr” and “CMP masked 6s plasma-Gr” systems are almost the same, except for the pores on the graphene. Then Zeo++ was employed to analyze the PSD of the membranes after being masked by the same CMP film.

It is essential to highlight that, for a more comprehensive exploration of the interaction between the CMP polymer and porous graphene, a distinct strategy was used in these simulations compared to that

detailed in the main text. Instead of measuring the PSD of the entire membrane, comprising a 6-nm-thick CMP layer and a porous graphene layer, the focus was on investigating the interface between the CMP layer and the graphene layer. Specifically, a 6-nm-thick CMP layer was designed atop the porous graphene, and the CMP polymer was “sliced” at a distance of 1 nm from the graphene layer with the slicing direction parallel to the plane of graphene. Subsequently, a statistical analysis of the PSD of the porous graphene (encompassing both 5CO₂-Gr and 6s plasma-Gr) with the 1-nm-thick CMP polymer, closest to the graphene layer, was conducted.

Fig. S33 Simulated structures and PSD: a) front view of the porous graphene (5minCO₂-Gr) with different size pores, b) side view of the graphene membrane with a CMP mask layer, c) simulated PSD of the CMP-masked Gr membrane and the sizes of the pores in the 5minCO₂-Gr before masking, d) a partially enlarged image of Fig. c), e) front view of the 6s plasma-treated graphene (6s plasma-Gr) with four large pores, f) side view of the membrane with a CMP mask layer, g) simulated PSD of the CMP-masked 6s Gr membrane and the sizes of the pores in the 6s plasma-Gr before masking, and h) a partially enlarged image of Fig. g).

The results depicted in Figs. S33a-d reveal that in the CMP-masked 5minCO₂-Gr membrane, denoted as "CMP-masked-Gr" in the main text, more than 95% of the pores possess a size smaller than 0.86 nm (corresponding to the hydrated diameter of Mg²⁺). This implies a significant limitation on the diffusion of divalent ions. Simultaneously, no pores exceeding 0.9 nm are recorded, suggesting that all large pores are effectively masked by the CMP layer and segmented into smaller ones. The defects in the porous graphene have been successfully repaired. As discussed in the main text, CMP polymer effectively partitions non-selective large pores into distinct smaller ones, thereby narrowing the PSD. This observation is reminiscent of the pore-in-pore strategy, wherein the interplay between the pores in the graphene and those in the CMP leads to an overall reduction in pore size. However, in comparison, for the 6s plasma-Gr sample (depicted in Figs. S33e-h), upon being masked by the 1-nm-thick CMP layer, the predominant feature in the PSD curve is the existence of pores surpassing 0.8 nm, although the larger pores exceeding 2 nm disappeared. Approximately 50% of the membrane pores lie within the range of 0.7 nm to 1.0 nm. Significantly, this aligns with the PSD observed in the CMP polymer itself, as detailed in the literature^[2]. This occurs due to the diminishing synergistic effect between the large

pores (exceeding 2 nm) in the graphene and those in the CMP polymer, leading to increased exposure of the pores inherent to the CMP polymer. Notably, these CMP pores are larger than the hydration diameter of Mg^{2+} . Consequently, in ion tests, the CMP-masked 6s plasma-Gr membrane exhibits no discernible Li^+/Mg^{2+} selectivity. To a certain extent, the CMP polymer is limited in its ability to further reduce the pores in the 6s plasma graphene, as this is dictated by the intrinsic pore size of the CMP material.

The simulation experiments conducted above underscore the importance of precisely controlling the pore size in graphene membranes. Although CMP effectively narrows the pores in porous graphene, the ion selectivity is derived from the synergistic interaction between CMP and graphene pores. In this investigation, the micropores within the CMP polymer itself demonstrate inefficacy in achieving selectivity between monovalent and divalent ions. Even in the case of the CMP-masked 6s plasma-treated graphene, efficient ion separation remains elusive, primarily attributed to the excessively large size of the graphene pores.

Minor typos:

SI note S1 – should be polydopamine instead of dopamine?

Author's Response: We used dopamine hydrochloride which polymerized in the CNT solution during the solution preparation process and modified the CNT.

Page 2, line 48 – The sentence “For example, a masking layer composed of a Å-scale aperture would reduce the selectivity-deteriorating rapid nonselective transport from nanoscale tears and cracks as well as nanometer-sized large nonselective pores in graphene” could be rephrased for better clarity.

Author's Response: We express our gratitude to the reviewer for the nice suggestion. The sentence has been rephrased in the revised manuscript as “For instance, a masking layer with Å-scale apertures would mitigate the rapid, non-selective transport arising from the nanoscale tears, cracks, and large nonselective pores in graphene, thereby enhancing selectivity.”

Page 3, line 64 – “resist the thickness” should be “limit the thickness”; Page 7, line 157 – “an noticeable” to “a noticeable”

Author's Response: We express our gratitude to the reviewer for dedicating the time to provide this meticulous review. We have corrected the grammatical errors. Additionally, we have meticulously reviewed the entire manuscript.

Page 10, line 235 – Consider rephrasing “staircase pattern” – presumably this is caused by digitization of the measurement?

Author's Response: Yes, we agree. The observed "staircase pattern" is ascribed to the remarkably slow transfer rate of Mg^{2+} , which nearly approaches the lower detection limit of the conductivity meter. This leads to repetitive up-and-down fluctuations in the data over a certain time. The exceedingly slow Mg^{2+} permeation rate indicates the effective hindrance of its permeation through the CMP-masked pores, consequently contributing to an enhanced Li^+/Mg^{2+} selectivity.

SI, page 6 – “the detector isused”

Author's Response: It has been corrected as suggested.

Fig. S2 – The caption does not make it clear what is depicted in the SEM images.

Author's Response: We express our gratitude once again to the reviewer for the meticulous review. The caption has been revised to provide more detail and clarity.

Reviewer 2

In this work, Zhou et al. prepared a monolayer graphene membrane on CNT support and demonstrated a strategy by conjugated microporous polymer to seal the defects. The prepared membrane performed well for $\text{Li}^+/\text{Mg}^{2+}$ separation and stayed stable for 3 weeks. The work is interesting and well written. However, some questions below need to be addressed.

Author's Response: We appreciate the reviewer's valuable and constructive comments, which helped in improving the quality of this manuscript.

Question 1:

The electrochemical polymerization process could easily damage single layer graphene. Is the graphene layer still there after CMP polymers are grown? The authors provided some indirect evidences by showing that standalone CMP membrane is nonselective. However, the argument could also be that CMP layer that is grown alone might be very thick, or it may show different charge properties.

Author's Response: The graphene layer is indeed intact after the electrochemical polymerization which is supported by the following two evidences.

1) We probed the gas permeance of both CMP-CNT and CMP-Gr-CNT membranes where graphene was intentionally used as-synthesized (pristine Gr) hosting only intrinsic vacancy defects (Fig. R2). The control sample of CMP-CNT membranes showed very high permeance around 50000 GPU for CO_2 . However, the CO_2 permeance of CMP-pristine Gr-CNT decreased to approximately 30 GPU on average parity to CO_2 permeance from intrinsic vacancy defects in graphene^[14, 15]. It suggests the graphene layer is still there after the electrochemical polymerization and maintains great structural integrity.

Fig. R2 CO_2 permeance of CMP-CNT and CMP-pristine graphene-CNT membranes.

2) When pore size in graphene is much larger, beyond the hydration shell diameter of ions, we do not observe ion-ion selectivity. For this, we prepared CMP-masked 6s plasma graphene membranes, the $\text{Li}^+/\text{Mg}^{2+}$ selectivity is lower than 3 (Fig. S37), due to the large pore size (~2.4 nm on average) of the 6s plasma-treated graphene. Therefore, in the absence of a graphene layer or if the porous graphene layer is extensively damaged during electrochemical polymerization, achieving high ion selectivity is not feasible.

Fig. S37 Ion sieving performance of the CMP-masked graphene (treated by 6-second plasma) membranes. The CMP masking layers were prepared under different scan rates (50, 200, and 300 mV/s) with 5 CV cycles.

To address the concerns raised by the reviewer on the role of CMP thickness, we tested the ion sieving performance of the CMP films (without porous graphene) grown alone with various thicknesses by tuning the electrochemical polymerization conditions (with scan rates of 50, 200, and 300 mV/s, respectively) (Fig. S35). All these CMP membranes showed negligible $\text{Li}^+/\text{Mg}^{2+}$ selectivity lower than 2 with various Li^+ permeation rates from $4.88 \pm 1.57 \times 10^{-5}$ to $4.12 \pm 1.03 \times 10^{-3}$ m/h. This result indicates that the lack of selectivity in the CMP film is not attributed to its thickness difference.

Fig. S35 Ion sieving performance of the CMP-CNT (without graphene) prepared under different scan rates (50, 200, and 300 mV/s) with 5 CV cycles.

Regarding the charge properties, the zeta potential of the CNT support, porous Gr, CMP membrane, and CMP-masked Gr have been checked and compared in the revised SI (Supplementary Note S11), and reproduced below:

“Regarding the charge properties, the zeta potential of the CNT support, porous Gr, CMP membrane, and CMP-masked Gr have been checked and compared. All samples showed negative surface charge from pH=3 to pH=10, which is due to the strong negative charge on CNT. It should be noted that the

CMP film itself is not charged, which was investigated and reported in the literature^[3]. The porous graphene prepared by CO₂ expansion at 800 °C is devoid of functional groups. Indeed, functional groups are not detected by XPS as shown in Fig. S17c. The reason for these membranes to be negatively charged would be due to the influence of the negatively charged polydopamine CNT substrate. Even though the porous CMP polymer has a thickness of ~10 nm, the effects of this substrate on the upper layer's potential have been widely reported^[16, 17].

More possible reasons and potential mechanisms are out of the scope of this work.

Fig. S31 Zeta potential of a) CNT support, b) porous Gr on CNT support, c) CMP on CNT support, and d) CMP-masked Gr on CNT support.

Additionally, we have tested the membranes under different pH values. The ion sieving performance tested under different pH values (4, 6.6, and 8) revealed that ion permeation rate and ion selectivity did not change significantly as a function of pH (Figure S39). More detailed explanations are provided in the response to Question 4.

Fig. S39 Ion sieving performance as a function of pH.

Hence, the lack of selectivity in the CMP layer itself is likely not due to its thickness or different charge properties, but rather attributed to its large pore size and material properties, which is also consistent with that in the literature^[2].

Question 2:

What is the pore size of the CMP polymer? The film is 10 nm thick, much thicker than single layer graphene. It's surprising that the layer does not cause resistance to ion transport. In fact, earlier, transport is largely contributed by the non-selective flow through the large pores in graphene. When pores are covered, the selectivity is much enhanced, while monovalent transport is not largely reduced. How does CMP contribute to the ion transport? The authors should provide more analysis.

Author's Response: The pore size of the CMP polymer is ~ 0.85-1.0 nm as reported by nitrogen-physorption isotherms in the literature^[2]. The hydration diameter cutoff (HDCO), associated with 90% ion rejection, is also reported based on which pore size is estimated to be 0.95 nm^[2]. The Brunauer–Emmett–Teller (BET) surface area of the polymer is 809 m²/g, which is comparable to that of COFs^[18], zeolite^[19, 20], and MOFs^[21], suggesting its highly porous structures.

It should be noted that although the CMP polymer is porous with intrinsic conjugated micropores, it still increases the resistance to ion transport. And the impact of this rise in ion transfer resistance is even more significant for larger-sized ions. Specifically, the average Li⁺ permeation rate decreased by a factor of 5, from 6.7×10^{-4} m/h to 1.3×10^{-4} m/h (the unit is normalized by the concentration following reviewer 1's suggestion) after the electrochemical repair.

Additionally, the ion transport resistance from the CMP layer increases with its thickness, as illustrated in Fig. S38. The CMP-masked Gr membranes with a CMP layer prepared under 3 CV cycles exhibited an average Li⁺ permeation rate of 2.7×10^{-4} m/h, while the membranes with a 10 CV-cycles CMP mask layer showed an order of magnitude lower average Li⁺ permeation rate around 3.1×10^{-5} m/h. Compared with the porous graphene, this represents a transport rate reduction by 2.5 times and 22 times, respectively.

Fig. S38 Ion sieving performance of the CMP-masked graphene (treated by 5-minute CO₂) membranes with various CMP layers prepared with 3, 5, and 10 CV cycles under the scan rate of 200 mV/s.

We believe that the lack of detailed discussions in the main text may contribute to confusion on this topic. We had a systematical discussion in the supporting Information regarding the effects of the CMP on the ion sieving performance (Supplementary Notes S13 and S14). However, to better communicate this point, we have now added the following sentences in the main text (Lines 266-268):

“Additional results about the optimization routine of the CMP layer and the impacts of CMP thickness on the ion sieving performance are presented and discussed in the Supporting Information (Supplementary Notes S13 and S14, Figures S35, S37, and S38).”

Question 3:

What is the performance of the membrane on the separation of other ion pairs in a binary system?

Author’s Response: In addition to testing the Li⁺/Mg²⁺ binary system for 500 hours, we have tested K⁺/Mg²⁺ and Na⁺/Ca²⁺ binary systems for 100 hours. The CMP-masked Gr membranes showed stable selectivity as shown in Fig. S42. The results have been added to the revised SI.

Fig. S42 Long-term a) K⁺/Mg²⁺ and b) Na⁺/Ca²⁺ binary ion sieving performance of the CMP-masked porous graphene membrane.

Question 4:

Will pH affect the separation?

Author's Response: We thank the reviewer for raising the constructive and helpful question. We have now tested the membranes under different pH values. The results have been now added to the main text (Lines 269-275)

“The ion sieving performance tested under different pH values (4, 6.6, and 8) revealed that ion permeation rate and ion selectivity did not change significantly as a function of pH (Figure S39). Although zeta potential experiments showed negative charges of the CMP-masked membranes, it is likely from the strong influence of the negatively charged CNT support as widely reported in the literature^[16,17] (Supplementary Note S11, Figure S31). We note that the pore size of the CNT support is excessively large (≥ 10 nm), and hence Donnan exclusion can be ruled out from the CNT support.”

Fig. S39 Ion sieving performance as a function of pH.

Question 5:

As the pore size of the membrane is around 0.4 nm, while the size of hydrated Cl⁻ is 0.664 nm, will the Cl⁻ be blocked by the membrane, and will it affect the cation separation?

Author's Response: The CMP-masked Gr membrane exhibited a pore size distribution ranging from 0.2 to 0.9 nm with a peak around 0.45 nm as shown in Fig. 3i. We note that partial ion dehydration is likely occurring in our system. Therefore, Cl⁻ ions are anticipated to permeate the membrane after partial dehydration. Notably, the hydrated sizes of K⁺ and Li⁺ ions are closely comparable to that of Cl⁻ ions. This enables the separation of these monovalent ions from much larger divalent ions.

Fig. 3i Simulated PSD of porous graphene with (top) and without CMP mask (bottom).

Question 6:

What is the mechanism of the membrane for cation separation?

Author's Response: We believe the membrane separates monovalent cations and bivalent cations based on a combination of effects including partial ion dehydration to enter the pore (where ease of dehydration plays a role) and steric hindrance (where size difference between the pore and the diameter of hydrated ions plays a role). It should be noted that porous graphene prepared by pore expansion at 800 °C is devoid of functional group (XPS as shown in Fig. S17c). Literature^[3] also establishes that CMP is not negatively charged. While CNT is negatively charged, its pore size is too large to influence ion rejection.

We further note that a small negative charge on graphene pores may not be ruled out due to the presence of residual oxygen functional group which is challenging to detect in XPS. However, in this case, as well, the small negative charge will not contribute significantly to the LiCl/MgCl₂ separations. Generally, the sieving performance of a charged membrane is governed by the co-ions (ions showing the same charge sign as the membrane)^[22]. A typical example is the successful separation of NaCl/MgSO₄ by the negatively charged polyamide TFC nanofiltration membranes due to the higher valence of SO₄²⁻ than that of Cl⁻. But for NaCl/MgCl₂ and LiCl/MgCl₂ separations (with the same co-ion, Cl⁻), polyamide TFC membranes do not show high performance^[23, 24]. This is because pores in polyamide are too large (or the pore size distribution is too broad) for the separation of Li⁺ and Mg²⁺.

Reviewer 3

This paper reports the fabrication of CMP-masked graphene membranes for ion separation. The methodology of creating and healing the graphene pores is interesting, with outperforming $\text{Li}^+/\text{Mg}^{2+}$ selectivity. However, the authors need to demonstrate a better understanding of the interaction between the CMP-masked layer and graphene pores. Moreover, the current ion sieving test through self-diffusion process is less applicable. This paper may be published in Nature Communications if the comments below are addressed.

Author's Response: We appreciate the reviewer's valuable constructive review. We have revised the manuscript and included additional results to address the reviewer's concerns.

Question 1:

1. The authors attributed the change of pore size to the stacking of CMP and graphene pores, but the key question is whether the CMP layer was only deposited on the graphene surface or intruded into the pores. To address this concern, the authors could carry out an XPS etching experiment to explore the in-depth profile of the CMP layer.

Author's Response: Following the reviewer's suggestion, we have done the XPS etching experiment. The results have been added and discussed in the revised SI, and also provided below for your convenience.

XPS depth profiling analysis provides valuable insights into membrane structures. However, determining the polymer's conditions (whether it deposits on the graphene surface or intrudes into the pores) through XPS is challenging due to the limitations in detection accuracy and the single atomic layer thickness of the graphene.

Based on the simulated results, our findings suggest that CMP polymer chains have the potential to infiltrate pores of larger size, as illustrated in Fig. S32c. However, it appears that intruding smaller micropores poses a greater challenge. More detailed work deserves to be conducted in the future.

Fig. S32 Simulated structures: a) front view of the porous graphene with different size pores, b) side view of the porous graphene, c) side view of the CMP-masked porous graphene membranes.

(Revised Supporting Information)

Supplementary Note S8

XPS depth-profiling analysis

To accurately detect the components near the graphene layer, we conducted a pre-experiment to determine the etching thickness via controlling the etching parameters (sputter time, accelerating voltage, current, and so on). A freestanding porous graphene was transferred to a silicon wafer coated with 100 nm thick Au metal. After the electropolymerization, the sample CMP-porous Gr-Au-Si was obtained. The graphene layer is attached to the Au metal layer, the etching conditions are determined once the Au element is detected. Next, we employed the same etching conditions, ensuring the detection of elements near the graphene layer, to etch the CMP-masked Gr membranes with a CNT support (CMP-porous Gr-CNT-PTFE). Finally, the sputter time was set to 0s, 50s, and 100s to detect the components on the surface of the sample, near the porous graphene layer, and the CNT support, respectively. After each etching, the XPS spectra of C and N were recorded for comparison.

The evolution of C1s and N1s peaks as a function of sputter time for the samples is shown in Fig. S28. After fitting, the C1s spectrum obtained from the sample surface (sputter time=0s) could be split into two peaks attributed to C-C/C-H/C=C at 284.8 eV and C-N/C-OH at 286.7 eV of the CMP polymer^[25]. After sputtering for 50 seconds, the C1s spectrum showed no significant changes, suggesting that the CMP polymer near the porous graphene layer is almost the same as the CMP on the surface. Further increasing the sputtering time to 100 seconds, a new peak at 289.0 eV was observed in the C1s spectrum, which should be attributed to N-C=O/O-C=O from the polydopamine covered on CNT^[26, 27].

The N1s spectrums as a function of sputter time are shown in Fig. S28d. The N1s spectrum obtained from the surface CMP (sputter time=0s) shows one main peak, which should be attributed to the R₂-NH group in CMP polymer^[28]. The characteristic R-NH₂ peak of dopamine at 401.4 eV is not observed in this spectrum. It indicates that a CMP layer uniformly covers the porous graphene-CNT layer. After etching for 50 seconds, the R-NH₂ peak was observed, which is associated with dopamine and polydopamine^[27]. The N1s spectrum obtained at 50 s is similar to that obtained with a sputtering time of 100 seconds. The latter reaches the depth of the CNT support layer. Although etching for 50 seconds should reach the depth of the graphene layer, it still collects signals from the underlying CNT layer. It is reasonable because the XPS detection signal originates from a depth of several nanometers from the sample surface, especially, given that graphene is only 0.34 nm in thickness. The XPS depth-profiling analysis helps confirm the multilayer structure of the membrane.

(Revised Supporting Information)

Supplementary Note S9

Membrane structure analysis

After a comprehensive set of characterizations, including XPS depth-profiling, TEM FIB cross-section, TEM, and SEM, we conclude that the membrane's structure is likely as follows (Fig. S29):

The CMP monomers will distribute on the surface and inside the CNT support of the membrane once the Gr-CNT membrane is in contact with the reaction solution. Next, under the action of an electric field, the CMP monomers become CMP polymer after a series of oxidation and reduction reactions. The reactions extended the polymer chain and reduced its solubility. Ultimately, a CMP film was deposited in situ on the conductive substrate. Since both graphene and CNT possess excellent conductivity, CMP grows and deposits on the surfaces of both Gr and CNT as shown by the following schematic. It is easier to form a continuous thin film by depositing it on the surface of the graphene layer (top side of the

membrane). Therefore, it is easy to observe a CMP polymer layer on this side under SEM. However, it needs more polymer to form a continuous film on the CNT side (bottom side), because of the relatively large space and specific surface area inside the CNT network support. On the CNT side, the polymer first covered the CNT bundles filled all the gaps between the CNT, and then formed a continuous thin film. Therefore, the morphologies of the two sides are very different, which is supported by the SEM (Fig. S25) results. Additionally, the structure of the membrane is also supported by TEM FIB and XPS results.

Based on the above analysis, we hypothesize that the polymer chains can intrude into the large pores of the graphene. Although the pores of graphene are empty (not conductive per se), the underneath CNT bundles are conductive. Additionally, given that the size of these pores is on the nanometer scale, the CMP polymer can still deposit or form on and across the pores in the graphene layer. However, current technological capabilities are insufficient to characterize it.

Fig. S28 XPS depth profile analysis. a) Schematic structures of the membrane, b) schematic structures for the XPS depth-profiling of the membrane, c) evolution of C1s peaks as a function of sputter time, and d) evolution of N1s peaks as a function of sputter time.

Fig. S29 a) Schematic structures of the membrane, b) schematic of CMP polymer covered CNT, and c) TEM image of CMP polymer covered CNT.

Question 2:

2. What is the surface charge of the membranes? Fig.S20 shows positive charged amines in the CMP backbones, which could significantly contribute to the high Li⁺/Mg²⁺ selectivity as proved by many reports in literature. This may not be pronounced for CMP films alone since their pore size was too loose to create Donnan effect, as shown in Table S2. However, if the CMP was intruded into graphene pores, the effect of charge could be more important than that of the pore size for achieving the high Li⁺/Mg²⁺ selectivity.

Author's Response: Please also see the response to reviewer 2 question 1, question 4 and question 6. Briefly, graphene and CMP in this study are not charged. The porous graphene is prepared by CO₂ expansion at 800 °C where most of the functional groups are lost. Indeed, functional groups on graphene are not detected by XPS as shown in Fig. S17c. We carried out zeta potential measurements (Fig. S31) which indicated that CNT support structure is charged which influenced the zeta potential results of CMP/CNT and CMP on Gr-CNT.

Regarding the neutral charge on CMP, we have clarified this further in Supplementary Note S11 (below):

Supplementary Note S11

Zeta potential analysis

Although there are positively charged amines in Fig. S22. They are carbazole cations, which are the

intermediate products of oxidation half-reactions. These positively charged cations undergo a reduction in the following reduction half-reactions. Additionally, the reductive peak appeared at 0.64 V in the negative scan, but the reductive current was set as low as -0.8 V, ensuring that the product was completely reduced to a neutral state. Therefore, from a chemical perspective, the CMP polymer cannot carry positive charges.

Regarding the charge properties, the zeta potential of the CNT support, porous Gr, CMP membrane, and CMP-masked Gr have been checked and compared. All samples showed negative surface charge from pH=3 to pH=10, which is due to the strong negative charge on CNT. It should be noted that the CMP film itself is not charged, which was investigated and reported in the literature^[3]. The porous graphene prepared by CO₂ expansion at 800 °C is devoid of functional groups. Indeed, functional groups are not detected by XPS as shown in Fig. S17c. The reason for these membranes to be negatively charged would be due to the influence of the negatively charged polydopamine CNT substrate. Even though the porous CMP polymer has a thickness of ~10 nm, the effects of this substrate on the upper layer's potential have been widely reported^[16, 17].

The negative charges on the CNT support layer are not expected to affect Li⁺/Mg²⁺ separations. This is mainly because the CNT support layer has a large pore size. This is also because in general, the sieving performance of a charged membrane is governed by the co-ions (ions showing the same charge sign as the membrane)^[22]. A typical example is the successful separation of NaCl/MgSO₄ by the negatively charged polyamide TFC nanofiltration membranes due to the higher valence of SO₄²⁻ than that of Cl⁻. But for NaCl/MgCl₂ and LiCl/MgCl₂ separations (with the same co-ion, Cl⁻), polyamide TFC membranes do not show high performance^[23, 24]. This is because pores in polyamide are too large (or the pore size distribution is too broad) for the separation of Li⁺ and Mg²⁺.

Question 3:

3. In Table S2, the ion concentration used in this paper was 0.1M, whereas most the comparative work used higher ion concentrations. This is not a fair comparison as concentration polarization could be more severe at the membrane surface for higher ion concentrations.

Author's Response: Based on the reviewer comments, we have normalized the performance by the concentration difference and the x-axis in the unit has been updated by (mol/m²/h)/(mol/m³) = m/h. Additionally, we also compared the performance of the membrane driven by different salt concentrations, And the results show that there is not a significant difference in normalized ion permeation rate. The results have been added in the revised Supporting Information.

(Revised Supporting Information)

Supplementary Note S14

Effects of concentration on ion sieving performance

The ion separation performance of the CMP-masked Gr membranes driven by different salt concentrations was compared (Fig. S40). The results show that there is not a significant difference in normalized ion permeation rate (with an average value of 5.03×10^{-4} for Li⁺ and 2.60×10^{-7} Mg²⁺) and ion selectivity tested under different concentrations. The observed phenomenon may be attributed to the vigorous stirring facilitated by a magnetic stirrer. Additionally, the large volume of the diffusion cell may play a role in mitigating the impacts of concentration polarization.

Fig. S40 a) Ion permeation rate as a function of salt solution concentration, b) ion permeation rate with unit normalized by the concentration difference.

Question 4:

4. The pristine graphene membranes in Fig. 1e showed good stability in pressure driven filtration up to 8 bar, why was the ion sieving experiment through CMP-masked graphene membranes not performed under the same conditions instead of self-diffusion? Self-diffusion is a less attractive process for the real industrial application. Also, the high Li⁺/Mg²⁺ selectivity in self-diffusion process may not reflect the actual selectivity in the real operation under high pressures.

Author's Response: We agree with the reviewer that self-diffusion is less attractive in industrial applications. While pressure-driven filtration is practically the most relevant to the intended applications, concentration-driven ion sieving is also commonly used to evaluate the membranes, especially for testing membranes that are made of novel materials or with ultrathin thickness. One of the major differences between filtration and self-diffusion is the presence of water flux. As an ion-sieving (solute-solute separation) membrane (rather than a typical nanofiltration membrane), CMP-masked Gr membranes showed a very fast ion permeation rate, but the water flux was very limited, restricting ion separations under pressure-driven filtration processes. This limitation could be attributed to the material properties of the CMP polymer.

An attractive application of the CMP-masked Gr membrane is flow-battery energy storage^[7], which is a topic of an ongoing project. The ion-sieving membranes transport ions while isolating the electrochemical reactions in separate cells. Effective ion separation under electric field-driven ion diffusion is of practical significance for the flow battery membranes. Here, we provide some results tested under an electric field to help address the reviewer's concern about the practical application of the membrane.

(Method section, lines 407-417)

Electric field -driven ion sieving tests

The Electric field-driven ion sieving tests were conducted using an electrochemical workstation (VSP-300, BioLogic) with an electrochemical cell. The membrane was placed between the two chambers of the cell, and each chamber was filled with a 10 mmol/L salt solution. Additionally, a pair of Ag/AgCl

electrodes were carefully positioned in the chambers. Ionic conductance was determined within the voltage range of -0.2 to 0.2 V, with a step size of 0.02 V/s, and the resulting current was recorded as a function of the applied voltage. The selectivity, S_e , is defined based on equation (5),

$$S_e = \frac{G_a}{G_b} = \frac{I_a/U_a}{I_b/U_b} \quad (5)$$

where G (S), I (A), and U (V) represent average electrical conductance, average current, and potential, respectively, and subscripts a and b represent ion a and b, respectively.

(Revised Supporting Information)

Supplementary Note S16

Ion sieving performance under electric field-driven process

The electric field-driven ion diffusion tests (Figs. S44 and S45) were employed to assess the ion separation performance of the CMP-masked Gr membranes. The results reveal that Li^+ ions displayed a significantly faster permeation rate due to their smaller size compared to the divalent ions Ca^{2+} and Mg^{2+} . The calculated conductance of Li^+ is $5.2 \pm 0.7 \times 10^{-2}$ mS, approximately 30 times higher than that of Mg^{2+} . It suggests the great potential in electric field-driven applications. This transport property aligns with observations from the concentration-driven permeation tests. It is noteworthy that the ion selectivity tested under the electric field-driven process may not be very close to that tested under the concentration-driven process. This discrepancy is likely explained by the forced migration of large ions due to partial dehydration when an electric field is applied, as reported in many studies^[29-31].

Fig. S44 Schematic of the setup for the ion sieving tests under electric field.

Fig. S45 Current-voltage (I - V) characteristics CMP-masked Gr membranes recorded in LiCl , CaCl_2 , and MgCl_2 solution.

Reviewer 4

The work entitled Electrochemical-repaired porous graphene membranes for precise ion-ion separation by Zhou et al. presents experimental and theoretical investigations on the ion separation by means of nanoporous graphene membranes. I believe that the covered topic is interesting for a broad community working in advanced filtration field.

The manuscript details a meticulous study of nanoporous graphene for ion-ion separation and the use conjugated microporous polymer to statistically reduce the pore size of nonselective larger pores. The nanoporous graphenes used in the current work are the same ones that were used in previous works (for instance Huang et al., *Adv. Mater.* 2022, 2206627 or Lee et al., *ACS Nano.* 16, 15382–15396, 2022). One half of the paper is reporting these previous results. This does raise questions regarding the novelty of the current manuscript. What mainly distinguishes this paper from previous works is that this time they used well-established support membrane to improve ion selectivity.

The work is very comprehensive, well carried out, and the scientific presentation is good. The authors certainly deserve credit for this study, and this paper should get. However, my concern with publishing this study in *Nature Communications* is that there is no real material novelty. The work seems to supplement the previous papers by the same authors. Therefore, I am hesitant to recommend publication in *Nature Communications*.

Anyway, here some issues that could clarify and potentially improve this draft paper.

Author's Response: We sincerely appreciate the reviewer's comment which allowed us to improve the presentation of this manuscript. We acknowledge that the lack of a detailed description of the experimental methodology may have led to concerns raised by the reviewer.

This work is fundamentally different from previously reported works as mentioned by the reviewer. The earlier studies highlighted by the authors do not focus on ion-ion separation. A comparison highlighting the differences, including materials, pore-forming methods, pore size, pore-size control methods, membrane structures, and applications, is summarized in Table R1 below. Briefly, The study by Huang et al. (*Adv. Mater.* 2022, 2206627) explores the pore formation mechanism on graphene through epoxidation and cluster gasification. This fundamental study teaches one to form narrow pores close to 3-3.3 Å for light gas separation. It does not deal with the pores (~ 7-8 Å) for ion separation. Lee et al. (*ACS Nano.* 16, 15382–15396, 2022) focus on the phase change of water from graphene pores placed at the liquid–vapor interface. However, the pores were generated by plasma treatment and the method of preparation of pores is not robust enough as the work.

We agree that porous graphene and electropolymerized polymers have been known. However, this report is the first one to combine these two to allow the realization of record-high performance. To our best knowledge, this is the first time that electrochemistry has been used for the preparation of graphene-related membranes. However, the preparation of the supported film is not trivial. Our initial attempt to directly mask porous graphene resting on Cu faced challenges from the side reaction of Cu. We spent significant efforts to arrive at a successful strategy for the precise deposition of electropolymerized film without inducing cracks or tears in graphene.

Therefore, the novelty of this work is the successful incorporation of electropolymerization of CMP (which is nontrivial) to act on graphene as a mask for its pores to repair pore size distribution while ensuring that cracks and tears do not form on graphene.

Table R1. Comparison between Huang’s work, Lee’s work, and this work.

	Pore-forming method	Effective pore	Pore size control method	Support Structures	Application
Huang et al. ^[32]	O ₃ oxidation + gasification at 100-200 °C	3.0 ~ 3.3 Å	Heating at 100-200 °C	Nanoporous carbon (NPC)	Gas separation
Lee et al. ^[33]	O ₂ plasma	3 Å	Plasma time	Nanoporous carbon (NPC)	Water evaporation
This work	O ₃ oxidation followed by pore expansion in CO ₂ at 800 °C	7.64 Å (Li ⁺) < size < 8.56 Å (Mg ²⁺)	electrochemical masking of pore	CMP/CNT	Ion-ion separation

Question 1:

The authors should clarify if the pore edges are hydrogenated or not.

Author’s Response: We do believe that the graphene pore edge is hydrogenated. This is mainly because pores are expanded in H₂/CO₂ atmosphere where most of the O-functional groups in the pores are lost. This is illustrated by XPS (Fig. S17). However, it is challenging to directly prove hydrogen termination of pore. We assume that when the oxygen functional group is lost, pores will get hydrogen terminated in H₂ environment at 800 °C. Analysis of XPS is described below:

The pristine graphene yields a C1s peak related to sp²-hybridized carbon (Fig. S17a). After the O₃ treatment, a new peak at 286.6 eV was observed (Fig. S17b), which is attributed to the epoxy group from the oxidation by O₃. After heating and subsequently CO₂-expansion at 800 °C, the O-functional groups on the pore edges were removed by the high-temperature reaction, partly by desorption and partly because these groups surround the pore and consequently are eliminated during expansion. It is supported by the disappeared epoxy peak as shown in Fig. S17c. This is consistent with our earlier work which characterizes the evolution of the O functional group as a function of temperature and shows that most functional groups are lost at 800 °C^[34]. Given the presence of H₂ in our expansion chemistry at 800 °C, we anticipate that pore edges will be hydrogenated.

Fig. S17 C1s XPS spectra of a) pristine graphene, b) O₃-treated graphene, and c) CO₂-treated graphene.

Question 2:

The authors should further elaborate on the pore diameter quantification. In figure 2d, the histogram displays bars of 0.02 nm. How they calculate the pore diameter? What's the associated error? Has sense to have around pores with a diameter of 0.02 nm?

Author's Response: We thank the reviewer for raising this important point. We have now incorporated the pore size quantification method into the revised SI for clarity.

The width of the bars in the histogram depends on how the data is presented (method of data counting and length of intervals). 0.02 nm (0.2 Å) is relevant for gas separation but likely not for monovalent/bivalent ion separation. Therefore, we revised the PSD (Fig. R3) where we redrew Figure 2d with histogram bars of 0.1 nm. We note that the results and conclusions did not change with the width of the bars.

(Revised Supporting Information)

Supplementary Note S5

Pore size quantification

The pore diameter is determined through the analysis of TEM images. After the image processing, a sphere fitting method is employed to determine the effective pore diameter. Specifically, a circle is fitted inside a pore by locating the outermost carbon atoms at the edge. The resulting diameter (D_i) is obtained as illustrated in Fig. S19. This circle contacts the outermost edge of the carbon atoms. Subsequently, adding the atomic radii of carbon atoms, we constructed another circle intersecting the center of these edge carbon atoms. The diameter of this circle is denoted as D_C . Next, we subtracted the nonbonded

interaction length between graphitic carbon and a water molecule, represented by $2^{1/6} \sigma$ ($\approx 3.86 \text{ \AA}$) corresponding to the position of the potential well^[1], from D_c . This subtraction yields an effective pore diameter. For example, a D_c value counted is 0.41 nm, the effective pore size after the subtraction is calculated as 0.024 nm ($0.41 - 0.386 = 0.024$), which is counted in the first bar of the histogram with the range of 0-0.025 nm.

By enumerating approximately 200 pores in the TEM images, we derived the pore size distribution for the sample. The pore density is computed by dividing the number of pores by the area of the selected TEM image.

Fig. S19 Schematic of the measurement of the pore size in the graphene.

Fig. R3 Schematic of the measurement of the pore size in the graphene.

Question 3:

Why pristine-Gr in figure 2f doesn't have error bars?

Author's Response: We regret that we did not display the error bars for the pristine-Gr membrane in the as-submitted manuscript which has been now resolved. The extremely low ion permeation rate of the pristine-Gr membrane was primarily utilized to demonstrate its structural integrity. Given that we used extensive characterization data (SEM, TEM), and ultralow water fluxes under pressure operation to demonstrate its structural integrity, we did not discuss the ion sieving performance of the pristine-Gr again here in Fig. 2f. We aimed to emphasize the performance of the 5minCO₂-Gr in this figure, leading

to the omission of error bars for the pristine-Gr membrane.

In response to the reviewer's recommendation, we have now incorporated the error bars to enhance clarity.

Fig. 2f Ion-ion separation performance of the membranes tested using different ions.

Question 4:

How many samples are studied in both configurations CNT-nanoporous graphene and CNT-nanoporous graphene- CMP mask? Figure 2e displays sets of 4 and 3 points, why? In figure 2e, are the four points indicated as Support four samples? Are those four samples the same ones indicated as Pristine-Gr, and then, as O3-Gr and so on? Are the CMP masked GR in figure 4d the same samples displayed in figure 2e?

Author's Response: We sincerely thank the reviewer for raising these important questions. We focused our efforts on generating data from at least 3 samples to understand any spread in the data.

Using the "Pristine-Gr" illustration from Figure 2e as an example, it comprises 4 data points, as highlighted by the reviewer. These data points represent four independent samples derived from four separate batches of synthesis. In each batch, the sample whose performance closely mirrors the average performance of all samples within that specific batch is considered representative. Our selection process is unbiased, as demonstrated by the Figure where the diversity of the data points for each type of membrane is relatively large. Typically, we present 3 representative samples; however, in cases where the standard deviation of a specific type of membrane's performance is significant, we include an additional data point to objectively illustrate the data. As shown in Figure 2e, the category "CMP-masked Gr" also contains 4 data points, and again, they are representative samples from 4 batches. And these 4 samples are not the same ones labeled as "Pristine-Gr".

Figure 4d serves as a summary of performance in this study, with comparisons among the membranes "Pristine-Gr," "Porous Gr," and "CMP-masked Gr" reported in previous figures in this study, as well as a comparison with literature results. Therefore, the data points in Figure 4d are the same as those found in Figure 2e and Figure 4b.

Question 5:

5. The authors claim that CMP-mask is playing a role as pore size reducer. Then, they should further

explain why figure S28 displays more than one order of magnitude in selectivity against the CMP mask thickness. One should expect a rough constant selectivity and a continuous increasing with increased thickness. A cross-section TEM image would be beneficial to further understand the CNT-nanoporous graphene – CMP mask interface for different CMP synthesis conditions.

Author's Response: Many thanks for the nice suggestion. Regarding Figure S28 (i.e. Fig. S38 in the revised version), we have included a detailed explanation of the effects of the CMP thickness on ion sieving performance in Supplementary Note S14. Briefly, the ion selectivity increases initially and then decreases with the thickness of the CMP layer. This is mainly because when the CMP layer is particularly thin or discontinuous, it cannot act as a mask layer to prevent the transfer of Mg^{2+} ions; when the CMP layer becomes particularly thick, mass transfer resistance increases, affecting the transfer of Li^+ ions as well, resulting in a decreased Li^+/Mg^{2+} selectivity.

We copy it here for the reviewer's convenience.

(Revised Supporting Information)

“Supplementary Note S14

Effects of the CMP thickness of CMP-masked Gr on the ion sieving performance

The effect of the thickness of the CMP mask layer on ion sieving was investigated (Fig. S38). The CMP layers with three different thicknesses prepared using 3, 5, and 10 CV cycles, at the scan rate of 200 mV/s, were used to mask porous graphene (5minCO₂-Gr). The estimated thickness of these CMP mask layers for the three CV cycles is 6, 10, and 20 nm, respectively. The obtained membranes were referred to as “3cCMP-masked Gr”, “5cCMP-masked Gr”, and “10cCMP-masked Gr”, respectively. 3cCMP-masked Gr exhibited a similar Li^+/Mg^{2+} selectivity to that of the porous Gr without CMP, around 17, suggesting negligible mask effects of the CMP layer to the large pores in the graphene. One potential explanation could be that the CMP layer obtained under 3 CV cycles is either excessively thin or discontinuous, rendering it inadequate in masking the large pores and defects in the graphene. Consequently, it cannot be successful in significantly enhancing ion selectivity as intended. On the other hand, 10cCMP-masked Gr showed a significant decrease in both ion selectivity and ion permeation rate. This suggests that an excessively dense CMP layer with significant thickness on porous graphene could impede the diffusion rate of Li^+ , primarily due to the substantial resistance encountered in mass transfer. As a result, both Li^+ ions and Mg^{2+} ions are unable to rapidly cross the membrane, resulting in the inability to achieve high selectivity. Ultimately, the CMP film prepared with 5 CV cycles under a scan rate of 200 mV/s was selected as the optimal mask layer.”

Preparation of a cross-section sample of an ultrathin membrane is challenging. Following the reviewer's suggestion, we tried TEM cross-section via the focused ion beam (FIB) technique. The cross-section TEM has been conducted and the results have been added to the revised SI. The results are also shown below for your convenience.

Fig. S27 a) and c) Cross-section TEM images of the CMP-masked Gr membrane, b) schematic structures of the membrane, d), e), and f) cross-section TEM images of the CMP-masked Gr membrane prepared under different CMP synthesis conditions.

(Characterizations section, lines 440-452)

“Focused ion beam (FIB) was conducted to check the cross-section of the samples. To achieve TEM cross-section images with enhanced contrast, the porous Gr-CNT membranes were transferred onto a silicon wafer coated with a 100-nm thick Au metal layer. Following the electrochemical repair process under varying conditions, the CMP-Gr-CNT-Au-Si samples were once again coated with a 100-nm thick layer of Au. Successively, from top to bottom, the TEM-FIB samples consist of six layers: An Au metal layer, a CMP mask layer, a porous graphene layer, a CNT support layer, a second Au metal layer, and the silicon wafer. To prepare the TEM lamella, a 1 μm-thick protective amorphous carbon film was deposited onto the sample using a 150 pA ion beam at 30 kV. Subsequently, a coarse milling and thinning procedure was executed, involving 6.5 nA for coarse milling and 3 nA/1.5 nA/0.7 nA for thinning. A final polishing step at low voltage (5 kV) and 30 pA for 10 seconds was conducted to eliminate carbon contamination generated during the FIB lamella preparation process. FIB lamella sample was imaged using FEI Talos F200s at an operating voltage of 80 kV.”

(Lines 227-240)

The TEM-FIB image depicts a well-defined cross-sectional cut, with individual layers in the stack clearly resolved and identified, aligning with our expectations. A uniform CMP layer is observed with a thickness of 14 ± 3 nm, consistent with the thickness measured by AFM. Although the single atomic layer thickness of the porous graphene layer is not discernible, the clear boundary between the CMP layer and the CNT support suggests the presence of a graphene barrier. No intrusion of CNT bundles

into the top CMP layer is observed. Moreover, the interface appears sharp and continuous, devoid of apparent gaps or voids, indicating a secure attachment between the CMP layer, porous graphene, and the CNT layer. The thickness of the CMP layer increases linearly with the number of cyclic voltammetry (CV) scans, in line with the AFM results. The interfaces between these CMP layers and the porous graphene layer show no significant difference. The CNT layer displays numerous voids and some cross-sectional structures of CNT bundles. The thickness of the CNT support measures 261 ± 11 nm, consistent well with the measurements obtained through SEM.

Question 6:

6. I suggest to perform Raman of the CMP-masked Gr to put some light into the effect of the electropolymerization on the nanoporous graphene.

In conclusion, I am inclined to not recommend for publication in Nature Communications journal.

Author's Response: The Raman has been performed and the results are shown below for your convenience. In response to the comments above, we have addressed the reviewer's comments on novelty and other questions and look forward to hearing back from the reviewer.

(Revised Supporting Information).

Supplementary Note S10

Raman spectrum of the CMP-masked Gr membrane

The Raman spectrum of the CMP-masked Gr membrane showed typical characteristics of carbon nanotubes with a D-band (1363 cm^{-1}), a G-band (1593 cm^{-1}), and a 2D-band (2717 cm^{-1}). Considering that the XPS depth profile analysis, SEM surface images, and TEM cross-section images all confirm the presence of a continuous CMP mask layer on top of porous graphene, we attribute this phenomenon to the influence of the underlying CNT support. To further investigate this hypothesis, we fabricated a thicker CMP polymer film with a thickness of ~ 20 nm and checked its Raman spectra. The result (Fig. S30) consistently aligns with that obtained from the CNT support, affirming our hypothesis.

Fig. S30 Raman spectra of CMP-masked Gr and CNT support.

References

1. Wu, Y. and N. Aluru, *Graphitic carbon–water nonbonded interaction parameters*. The Journal of Physical Chemistry B, 2013. **117**(29): p. 8802-8813.
2. Zhou, Z., et al., *Precise sub-angstrom ion separation using conjugated microporous polymer membranes*. Acs Nano, 2021. **15**(7): p. 11970-11980.
3. Zhou, Z., et al., *Flexible ionic conjugated microporous polymer membranes for fast and selective ion transport*. Advanced Functional Materials, 2022. **32**(6): p. 2108672.
4. He, G., et al., *High-permeance polymer-functionalized single-layer graphene membranes that surpass the postcombustion carbon capture target*. Energy & Environmental Science, 2019. **12**(11): p. 3305-3312.
5. Huang, T., et al., *Molecularly-porous ultrathin membranes for highly selective organic solvent nanofiltration*. Nature Communications, 2020. **11**(1): p. 5882.
6. Sarkisov, L., et al., *Materials informatics with PoreBlazer v4. 0 and the CSD MOF database*. Chemistry of Materials, 2020. **32**(23): p. 9849-9867.
7. Tan, R., et al., *Hydrophilic microporous membranes for selective ion separation and flow-battery energy storage*. Nature Materials, 2020. **19**(2): p. 195-202.
8. Zuo, P., et al., *Near-frictionless ion transport within triazine framework membranes*. Nature, 2023: p. 1-7.
9. Willems, T.F., et al., *Algorithms and tools for high-throughput geometry-based analysis of crystalline porous materials*. Microporous and Mesoporous Materials, 2012. **149**(1): p. 134-141.
10. Pinheiro, M., et al., *Characterization and comparison of pore landscapes in crystalline porous materials*. Journal of Molecular Graphics and Modelling, 2013. **44**: p. 208-219.
11. Hess, B., et al., *GROMACS 4: algorithms for highly efficient, load-balanced, and scalable molecular simulation*. Journal of chemical theory and computation, 2008. **4**(3): p. 435-447.
12. Oostenbrink, C., et al., *A biomolecular force field based on the free enthalpy of hydration and solvation: the GROMOS force-field parameter sets 53A5 and 53A6*. Journal of computational chemistry, 2004. **25**(13): p. 1656-1676.
13. Schüttelkopf, A.W. and D.M. Van Aalten, *PRODRG: a tool for high-throughput crystallography of protein–ligand complexes*. Acta Crystallographica Section D: Biological Crystallography, 2004. **60**(8): p. 1355-1363.
14. Huang, S., et al., *Single-layer graphene membranes by crack-free transfer for gas mixture separation*. Nature communications, 2018. **9**(1): p. 2632.
15. Rezaei, M., et al., *Hydrogen-sieving single-layer graphene membranes obtained by crystallographic and morphological optimization of catalytic copper foil*. Journal of Membrane Science, 2020. **612**: p. 118406.
16. Li, J., et al., *Fabrication and characterization of carbon nanotubes-based porous composite forward osmosis membrane: Flux performance, separation mechanism, and potential application*. Journal of Membrane Science, 2020. **604**: p. 118050.
17. Deng, L., et al., *Towards enhanced antifouling and flux performances of thin-film composite forward osmosis membrane via constructing a sandwich-like carbon nanotubes-coated support*. Desalination, 2020. **479**: p. 114311.
18. Shinde, D.B., et al., *Crystalline 2D covalent organic framework membranes for high-flux organic solvent nanofiltration*. Journal of the American Chemical Society, 2018. **140**(43): p. 14342-14349.
19. Dey, K.P., et al., *Preparation of NaA zeolite membranes using poly (ethyleneimine) as buffer layer,*

- and study of their permeation behavior. *Journal of the American Ceramic Society*, 2013. **96**(1): p. 68-72.
20. Bae, Y.-S., A.O.z.r. Yazaydin, and R.Q. Snurr, *Evaluation of the BET method for determining surface areas of MOFs and zeolites that contain ultra-micropores*. *Langmuir*, 2010. **26**(8): p. 5475-5483.
 21. Furukawa, H., et al., *Ultrahigh porosity in metal-organic frameworks*. *Science*, 2010. **329**(5990): p. 424-428.
 22. Tang, C. and M.L. Bruening, *Ion separations with membranes*. *Journal of Polymer Science*, 2020. **58**(20): p. 2831-2856.
 23. Liu, S., et al., *Tröger's base-regulated interfacial polymerization of polyamide nanofiltration membranes with enhanced performance*. *Journal of Membrane Science*, 2023: p. 121787.
 24. Dong, D., et al., *"Clickable" Interfacial Polymerization of Polythioether Ultrathin Membranes for Ion Separation*. *Macromolecules*, 2023. **56**(17): p. 7132-7141.
 25. Suárez-García, S., et al., *Copolymerization of a catechol and a diamine as a versatile polydopamine-like platform for surface functionalization: The case of a hydrophobic coating*. *Biomimetics*, 2017. **2**(4): p. 22.
 26. Zhu, Y., et al., *Single-walled carbon nanotube film supported nanofiltration membrane with a nearly 10 nm thick polyamide selective layer for high-flux and high-rejection desalination*. *Small*, 2016. **12**(36): p. 5034-5041.
 27. Batul, R., et al., *Polydopamine nanosphere with in-situ loaded gentamicin and its antimicrobial activity*. *Molecules*, 2020. **25**(9): p. 2090.
 28. Zangmeister, R.A., T.A. Morris, and M.J. Tarlov, *Characterization of polydopamine thin films deposited at short times by autoxidation of dopamine*. *Langmuir*, 2013. **29**(27): p. 8619-8628.
 29. Zhou, J., et al., *Highly conductive and vanadium sieving Microporous Tröger's Base Membranes for vanadium redox flow battery*. *Journal of Membrane Science*, 2021. **620**: p. 118832.
 30. Zhang, H., et al., *Ultrafast selective transport of alkali metal ions in metal organic frameworks with subnanometer pores*. *Science advances*, 2018. **4**(2): p. eaaq0066.
 31. Li, X., et al., *Fast and selective fluoride ion conduction in sub-1-nanometer metal-organic framework channels*. *Nature communications*, 2019. **10**(1): p. 2490.
 32. Huang, S., et al., *In Situ Nucleation-Decoupled and Site-Specific Incorporation of Å-Scale Pores in Graphene Via Epoxidation*. *Advanced Materials*, 2022. **34**(51): p. 2206627.
 33. Lee, W.-C., et al., *Enhanced water evaporation from Å-scale graphene nanopores*. *ACS nano*, 2022. **16**(9): p. 15382-15396.
 34. Bondaz, L., et al., *Selective Photonic Gasification of Strained Oxygen Clusters on Graphene for Tuning Pore Size in the Å Regime*. *Jacs Au*, 2023. **3**(10): p. 2844-2854.

REVIEWERS' COMMENTS

Reviewer #1 (Remarks to the Author):

The authors have addressed the reviewer comments in detail with additional experiments. The reviewer does not have further comments and can recommend the manuscript for publication.

Reviewer #2 (Remarks to the Author):

The authors have addressed my concerns.

Reviewer #3 (Remarks to the Author):

The additional experiments that the authors carried out have addressed all my concerns, especially for using these membranes in the applicable battery process. Therefore, I would recommend for publication in Nature Communications.

Reviewer #4 (Remarks to the Author):

The revised version of the paper entitled Electrochemical-repaired porous graphene membranes for precise ion-ion separation by Zhou et al. clearly answers the questions and includes a more detailed analysis of the required issues.

In conclusion, I am inclined to recommend the paper for publication in Nature Communications.